



# Timescales of C turnover in soils with mixed crystalline mineralogies, Kruger National Park, South Africa

Lesego Khomo[1], Susan Trumbore[1,2,], Carleton R. Bern[3], Oliver A. Chadwick[4]

[1]Max Plank Institute for Biogeochemistry, Germany
[2]Department of Earth System Science, University of California, Irvine, USA
[3]U.S. Geological Survey, Denver, CO, USA
[4]Department of Geography, University of California, Santa Barbara, USA

*Correspondence to*: Susan Trumbore (trumbore@bgc-jena.mpg.de)

**Abstract.** Organic matter-mineral associations stabilize much of the carbon stored globally in soils.   Metastable short-range-order (SRO) minerals such as allophane and ferrihydrite provide one mechanism for long-term stabilization of organic matter in soil.  However, ancient and highly weathered soils that cover a large fraction of land area lack SRO minerals. Here we evaluate the role of different minerals on the amount and turnover time (TT) of carbon in a field setting designed to minimize the role of SRO by taking advantage of multiple lithologies in Kruger National Park, South Africa. Density separation demonstrated that most of the C was associated with minerals, even in surface soils.  A parallel separation of clay-sized material demonstrated that 9-47% of the organic C in these soils was stabilized by clays. Organic C associated with clay-sized material had average TT of 1020 ± 460 years in surface soils. The mean TT of this clay-associated C increased with depth and with fraction of total clay that was smectite. Because the C associated with smectite clay was so old, the amount of smectite (2:1 clays) controlled the age of bulk soil C across Kruger landscapes.  The TT of the majority of soil C – not stabilized by clays - was much shorter, 190±190 years in surface horizons. We suggest that this faster component reflects timescales of weaker C stabilization by crystalline Fe and Al oxyhydr)oxides and kaolinite (1:1) clays, as well as LF fractions not associated with minerals.  Thus, bulk or HF carbon integrates C stabilized by mechanisms with inherently different TT, something that is often inferred from radiocarbon measurements.  While SRO mineral concentrations were very low in these soils, the soils with most SRO had very high C content but also very young C.  In other environments, SRO can be very stable and sorb C on very long timescales.  We hypothesize that the seasonal wetting and drying in the KNP may reduce the age of SRO minerals as well as the C associated with them. Across the varying lithologies and a precipitation gradient found in the KNP, we found mineralogy to be the most important explanatory factor for C content (related to crystalline Fe) and turnover time (related to the amount of smectite).



# 1 Introduction

The amount and persistence of organic carbon (C) stored in soils largely reflects interactions between organic matter and mineral surfaces (Lehmann and Kleber, 2015), although other mechanisms such as physical protection and preservation via recalcitrance play roles as well (Oades, 1984; Kaiser and Guggenberger, 2000; Kleber et al. 2007; Kögel-Knabner et al.,

2008). Carbon recovered in the dense fraction of soil can be stabilized by a range of interactions with inorganic compounds, but in the complex soil medium our understanding of the contributions of each mechanism is limited (von Lützow et al., 2007). For example, organic ligands can bond with trivalent metal ions such as iron (Fe) and aluminum (Al) or by dehydration bonding with metallic nano-(oxyhydr)oxides (Parfitt and Childs, 1988; Kaiser and Zech, 1996; Chorover et al., 2004). At high clay concentrations and soil surface area, organic ligands can be sorbed into the matrix of short-range-order

(SRO) nanocrystalline aluminosilicates or metallic (oxyhydr)oxides (Torn et al., 1997; Kaiser and Guggenberger, 2003; Chorover et al., 2004; Kramer et al., 2012) or onto crystalline silicate clay minerals (Kaiser and Guggenberger, 2000; Sollins et al., 2009). However 1:1 silicate clays with inherently low surface area, such as kaolinite and halloysite, have limited sorptive capacity (Heckman et al., 2009; Sollins et al. 2009). In contrast, 2:1 clays with high charge density and high surface area, such as smectite, have higher affinity for C that lowers rates of C turnover (Wattel-Koekkoek et al., 2003; Poch et al.,

2015). In addition, crystalline pedogenic Fe and Al sesquioxides are associated with C that has ages ranging from hundreds to thousands of years (Masiello et al., 2004).

Soil C can be operationally separated into a light fraction not associated with minerals (low density fraction; LF) and a heavy fraction associated with minerals (high density fraction; HF). Decomposition of the light fraction may be governed by interactions among decomposers and C substrate quality, as well as by occlusion within mineral aggregates. Stabilization

mechanisms in HF can operate on a range of timescales (e.g. Schrumpf and Kaiser 2014; Schrumpf et al. 2013; Wattel-Koekkoek and Buurman 2004).  Given the different strengths of mineral-C interactions, there is a need to measure carbon storage and age of C within the dense fraction of mixed mineralogy soils where the quantity of C-sorbing compounds including clay minerals has also been measured.  With this information, we can attempt to predict C storage and timescales of stabilization from known amounts of mineral constituents (e.g. 2:1 versus 1:1 clays) or from readily available proxy

measurements such as surface area or cation exchange capacity (Lawrence et al., 2015).

One approach to evaluate mineral controls on C storage and turnover is to select samples for analysis from distinctly different global environments to cover a suite of mineral compositions. For example, Wattel-Koekkoek et al. (2003) used this approach to document that larger quantities of crystalline 2:1 clays were associated with greater (and older) C storage, while 1:1 clays were not. Another approach is to sample soils along gradients of state factors to isolate specific soil property

gradients and evaluate correlations among varying soil properties and C properties. The latter approach has proved useful (e.g. Trumbore et al., 1996; Torn et al., 1997; Masiello et al., 2004) but often without full quantification of all the possible controls on C storage. By contrast Lawrence et al. (2015) quantified virtually all reactive components along a humid, forested chronosequence formed on alluvium from andesitic volcanics and evaluated their correlations with C storage and



turnover. They found a complex suite of controls for C storage and age related to depth-dependant C inputs and depth-dependant inorganic chemical properties. In surface horizons, they found increases in C storage but not its stability with increasing amounts of pyrophosphate extractable Fe and Al. In subsurface horizons they found increased C stability, but not increased storage, with increases in surface area and halloysite clay concentrations. In contrast with other studies cited above, they found little correlation between C storage or stability and the amount of smectite clays or SRO minerals. Although it would be ideal to be able to make predictions about C storage and stability based on chemical and mineralogical properties, this (Lawrence et al. 2015) and other studies (e.g. Masiello et al., 2004; Wattel-Koekoek et al., 2003; Wattel-Koekoek et al., 2004) indicate that there may be significant C-storage trade-offs occurring within soils of differing mineral compositions. Therefore we need more case studies that cover a broad range of environmental, ecological and soil chemical conditions. Our long-term goal is to refine understanding of the sensitivity of mineral-associated organic matter to climate and land-use change.

Few studies have focused on the C stabilization behaviour of mature soils where long-term depletion of primary minerals and ripening of secondary minerals provides an environment dominated by well crystallized compounds that have relatively low chemical reactivity (c.f. Wattel-Koekoek et al., 2003; Torn et al., 1997). In regions with long-term tectonic and climatic stability such as parts of the tropics and subtropics (Paton et al., 1995), it is possible that the differences in C sorption between 2:1 and 1:1 clays could be one of the most important controls on C storage and turnover.

Here we analyse a lithosequence of arid to subhumid savanna soils developed on the Kaapval Craton and associated post-Gondwana breakup lavas in Kruger National Park (KNP) South Africa (SA). Low rates of landscape erosion and exceptionally long soil residence times (Chadwick et al., 2013) ensure that nearly all soil minerals have evolved past the metastable SRO stage and that there are few free trivalent metal ions available for direct sorption by organic ligands (Khomo et al., 2011; Khomo et al., 2013). We evaluate $^{14}$C in bulk soil as well as clay-sized material, and LF and HF fractions and in particular the role of silicate clay mineralogy, particularly smectite (2:1) and kaolinite (1:1) clays, and non-silicate crystalline and non-crystalline Fe and Al sesquioxides in determining $^{13}$C and $^{14}$C, and estimates of long-term turnover of C in soils.

## 2 Materials and Methods

### 2.1 Field sites

To evaluate mineralogical controls on C storage and turnover we sampled soils across gradients in parent material, climate and topography in KNP. The KNP is a savanna between the SA border with Mozambique and the Great Escarpment. Soil residence times are >105 yrs (Chadwick et al., 2013), providing plenty of time for crystalline mineral differentiation and depletion of metastable SRO minerals. In addition to the strong lithological differences across KNP, secondary mineralogy heterogeneity is imposed by a regional north-south gradient in rainfall that ranges from about 470 to 740 mm annually, and locally by differentiation that varies soil clay content along hillslopes. Across these landscapes, we can focus on organic



matter – mineral interactions associated with differences in silicate clays and secondary Fe and Al (oxyhydr)oxides in a setting expected to have few SRO's.

We sampled soils underlain by five geological units: rhyolite, granite, picrite (olivine-rich or b1ack basalt), olivine-poor basalt (red basalt), and nephelinite (Venter et al., 2003) (Table 1, Figure 1). Each of the lithologies were sampled in the

northern, arid zone, with mean annual temperature of 23 °C and ~470 mm annual precipitation. We also sampled soils developed on granite, gabbro and mixed granite/gabbro parent materials in the southern quarter of the park where rainfall ranges from ~550 – 740 mm per year (Table 1). Samples were collected along watershed divides, i.e. hill crests in the gently rolling landscape, although we include data for soils collected along one toposequence at 550 mm of rain to increase the amount of mineralogical differentiation that develops along granitic catenas in the KNP (Khomo et al, 2011; Khomo et al.

2013, Bern et al., 2011).

Vegetation across the entire park is mixed grass-tree savanna. In the northern part of the park the dominant tree species across all lithologies is *Colospermum mopane*, with greater grass coverage on the more basic igneous rocks and greater amounts of woody shrub sub-dominants on the more acid igneous rocks (granite, rhyolite) (Scholes et al., 2003). The vegetation growth forms interact with patterns of fire frequency and herbivory, with greater fire frequency and heavier

grazing occurring on the grass-rich soils formed on the more basic igneous rocks (Scholes et al., 2003; Scholes and Walker, 1993). Vegetation at sampling localities on granite and gabbro parent materials in the southern (wetter) parts of the park consisted of grasses mixed with *Terminalia sericea* and *Sclerocarya birrea* (Scholes et al., 2003).

## 2.2  Soil Characterization

Soil samples were collected between 2004 and 2011. Profiles were sampled by horizon to bedrock where possible and

described and classified using standard techniques. Soil depth ranged from 30 cm to about 2 m with a minimum of three horizons and a maximum of eleven. Following air-drying, the samples were sieved to < 2 mm to remove rocks and roots. Air-dried samples were homogenized and sub-sampled for physical, chemical, isotopic and mineralogical analyses. Bulk density was measured as the mass of oven-dry soil in a core of known volume. Clay content was determined by the hydrometer method (Soil Survey Staff, 2014). The concentration of exchangeable base cations was determined by atomic

absorption spectroscopy after extraction with 1M ammonium acetate buffered at pH 7. Cation exchange capacity (CEC) was determined by extracting the ammonium saturated samples with a 1 M potassium chloride solution and determining ammonium by Lachat autoanalyzer. We report CEC corrected for the contribution of organic matter by assuming a contribution of 200 cmol(+) per kg organic C (as measured using an elemental analyser; Soil Survey Staff, 2014). Acid ammonium oxalate (AAO) was used to extract SRO Fe and Al in the dark (Schwertmann, 1973), while crystalline Fe and Al

were determined with a dithionite citrate bicarbonate (DCB) extraction (Mehra and Jackson, 1960). Iron and Al from both extractions were then measured by Inductively Coupled Plasms-Optical Emission Spectrometry. Carbon and nitrogen content were determined by combustion on a vario Max CN elemental analyser at the Max-Planck Institute for Biogeochemistry. For sites with pedogenic soil carbonates, inorganic carbon was determined on the residue after dry




combustion of bulk samples at 450 oC for 16 h (Steinbeiss et al., 2008) and organic carbon was calculated as the difference between total carbon and inorganic carbon.

## 2.3 Characterization of clay minerals

Sands were excluded by wet-sieving and the clays extracted following dispersion with 5% sodium hexametaphosphate and 2% $H_2O_2$, with three rounds of sedimentation and decantation in a 1L cylinder (Soil Survey Staff, 2014). The suspended clay was recovered by evaporation and freeze-drying. For measurement of C associated with clay minerals, we assume the H2O2 largely removed LF material (bubbling was observed in some samples) while leaving clay-associated C intact. We acknowledge that the 2% $H_2O_2$ treatment may affect the radiocarbon measurements of clay-associated materials, but note that the clays after this treatment still have C contents >1% (up to 4.5% C). The C and C isotope content of the non-clay material (which includes LF material oxidized by $H_2O_2$) was estimated from mass balance of the bulk and clay fractions. Separates of clay were saturated with KCl and $MgCl_2$ and rinsed until no $Cl^-$ ions were detected with one drop of $AgNO_3$. X-ray diffraction (XRD) analyses were conducted at the U.S. Geological Survey in Denver, Colorado. Peel-mounts of oriented clays were made by transferring clays onto microprobe glass slides from 0.42 μm cellulose nitrate membrane filters where they had been oriented by vacuum (Pollastro, 1982). For mineral quantification, the clay-size fraction was micronized in methanol with 10% $Al_2O_3$ by sample weight, dried, passed through a 50-μm sieve and placed into side-packed powder mounts (Eberl, 2003). XRD spectra were generated with a Siemens D500 diffractometer using Cu Kα radiation fitted with a graphite monochrometer configured to 35 mA and 40 kV. Mineral quantification was done using the Rockjock (Eberl, 2003) and results were summed by mineral group. Additional clay mineralogy data for select samples come from Khomo et al. (2011) and all mineralogy data are normalized to sum to 100%.

## 2.4 Density Separation

In addition to separating clay, we also performed a density separation on a separate aliquot of the bulk soil. Experience has shown the greatest contribution from LF to bulk soil C in A horizons; in deeper horizons, we did not perform density separations and instead assumed that the bulk fraction approximates HF for properties such as C and C isotope concentrations. We used a heavy sodium polytungstate liquid (1.7 g cm-3) to separate the LF and HF (Schrumpf et al. 2013). Approximately 10 to 15 g of soil was added to 100 ml sodium polytungstate solution and gently shaken on a horizontal shaker for 10 min, ultrasonicated at 60 J ml-1 for 2.5 min, then centrifuged at 3500 rpm for 30 min. The floating LF was concentrated on filter paper (1.6 μm glass microfiber discs) using a light vacuum. The sinking material was re-suspended in the heavy solution, and the steps repeated without the ultrasonic disaggregation until no more floating LF was observed (usually three times). The LF was rinsed with a litre of water to remove the heavy liquid, then freeze dried. Visible roots were removed by hand, and the remaining LF was ground to homogenize it for C isotope measurement. The HF was suspended in water, centrifuged and the solution decanted three times to remove heavy solvent, then dried and ground for C isotope measurement.





Thus our analyses of C and radiocarbon in fractions contain overlapping information. For example, we can assume that all of the C and $^{14}$C found in the clay fraction is also found in the HF fraction. The "non-clay" fraction, which is calculated using mass balance of $^{13}$C, $^{14}$C and C using data from bulk and clay fractions, contains a mixture of C associated with non-clay sized minerals (e.g. a proportion of the HF) as well as LF C (see also Results, Figure 4).

## 2.5 Carbon isotopes

Radiocarbon ($^{14}$C) was determined by accelerator mass spectrometry (AMS). For determination of $^{14}$C in organic samples, an amount of material (bulk soil, HF, LF, or clay) needed to yield ~1 mg C was weighed into a pre-combusted quartz tube with CuO wire. The tube was evacuated, sealed with a torch and placed in a 900 °C furnace for 3 hours. The resulting $CO_2$ was purified on a vacuum line, and an aliquot was removed for determination of $^{13}CO_2$ using a gas bench coupled to an isotope ratio mass spectrometer (Xu et al. 2007). The remaining $CO_2$ was reduced to graphite using a sealed tube zinc reduction method (Xu et al. 2007), and isotopic compositions were measured at the WM Keck Carbon Cycle AMS facility at the University of California, Irvine. Samples which contained inorganic carbon were acidified with 1N HCl until the solution pH was below 6 and then dried and analyzed as above. For one soil, we analyzed $^{14}$C in pedogenic carbonates by collecting and purifying the $CO_2$ evolved during acidification, then reducing it to graphite as for organic C samples.

Radiocarbon data are reported as $\Delta^{14}$C, the deviation, in parts per thousand, between the ratio of $^{14}$C /$^{12}$C in the sample divided by that of preindustrial wood (the standard) and 1. The potential influence of mass-dependent fractionation of isotopes is accounted for by reporting the $^{14}$C /$^{12}$C ratio corrected to a common $\delta^{13}$C value (-25‰), and assuming that $^{14}$C is fractionated twice as much as $^{13}$C by mass-dependent processes (Stuiver and Polach 1977). Therefore, differences in $\Delta^{14}$C between samples reflect time or mixing rather than isotope fractionation. In these units, $\Delta^{14}$C = 0‰ is equivalent to the standard. Values >0‰ indicate the presence of $^{14}$C produced by atmospheric thermonuclear weapons testing in the early 1960s. Values <0‰ indicate that radiocarbon has had time to radioactively decay (half-life = 5730 years). Long-term accuracy for samples measured at the WM Keck CCAMS facility is ±3‰ for radiocarbon expressed as $\Delta^{14}$C and ±0.1‰ for $\delta^{13}$C.

We used the radiocarbon data to estimate the mean turnover time (TT) of soil C in the profile using a one-pool model that includes incorporation of bomb-$^{14}$C in the last decades and assumes steady state (see Torn et al. 2007; Trumbore 2009). Specifically, we used the SoilR package in R (Sierra et al. 2014) to calculate the predicted radiocarbon signature for such a one pool, steady state model in the year of sampling (R code included in Supplemental Material). For cases where two turnover times yielded the same $^{14}$C in the year of sampling (for example, where $\Delta^{14}$C is >0‰), we report both TT for the LF fraction, but only the longer turnover time as more consistent with the fluxes of C into and out of the mineral associated and bulk fractions (see Gaudinski et al. 2000). TT are rounded to the nearest year (for <30 years) or the nearest 5 years (<1000 years) or nearest decade (>1000 years). The one-pool model is clearly an oversimplification, but is useful for translating radiocarbon data into average timescales of stabilization. The use of a mean TT also provides a way to compare data from samples collected in different years (2004–2011; Table 1).



We report carbon concentration and isotope data for individual horizons as well as whole-profile averages (e.g. as in Masiello et al., 2004). Mean carbon isotope ratios and mean estimated turnover times for whole profiles were calculated as averages, carbon-mass-weighted by horizon and calculated from measured bulk soil $^{13}$C and $^{14}$C values.

### 2.6 Statistics

Graphs, including regression analyses, were produced with R (R Core Team, 2015). Correlation matrices (see Suppl. Table 2) were produced using the R package Hmisc (Harrell et al. 2016).

## 3 Results

### 3.1 Mineralogy

With few exceptions, these ancient soils contained low amounts (<0.3%) of oxalate extractable Al+Fe presumed to be derived from SRO minerals (complete data are given in Supplemental Table 1). Only soils developed on the nephelinite (>0.5%) and the gabbro under sub-humid rainfall had greater (1.4%) concentrations of oxalate extractable Al+Fe. Crystalline Fe phases extracted with dithionite-citrate-bicarbonate (Fe$_d$) ranged from 0% in a periodically anoxic "seep" zone in the granitic toposequence to 4.6% in the nephelinite-soil. Dithionite-extractable Al exceeded 0.4% only in the subhumid

gabbro soil (2.1%; Suppl. Table 1).

The clay-size fraction made up ≤15% of the <2-mm mass for soils developed at crest positions on granites and rhyolites, but up to 35 - 50% in the red and black basalt, low rainfall gabbro and nephelinite soils, and at the toeslope of the granitic toposequence (Table 2; Suppl. Table 1). With a few exceptions, clay content increased with soil depth.

Within the clay-size fraction, the sum of smectite, kaolins, micas and chlorite and crystalline Fe minerals generally made up

over 90% of the quantified mineralogy (Table 2). Smectite was present in all of the isolated clays except the granite crest under relatively high (740 mm) rainfall, and dominated the clay fraction in basalt (>90%) and picrite (>99%; Table 2). Kaolin minerals were common in most soils but rare in the arid zone gabbro, picrite and basalt. Crystalline Fe oxide minerals identified by X-ray diffraction made up 3-26% of the clay-sized fraction for most soils, but <1% in the smectite-dominated basalt and picrite soils (Table 2).

### 3.2 C in density fractions (A and B1 horizons)

The LF isolated from A horizons had C concentrations of 10–37% (Table 3), with lower concentrations in the soils derived from extrusive igneous rocks (rhyolite and basalts; 10–16% C). In general, LF-C made up only 10–20% of the total bulk C,

even in surface soils (Suppl. Table 1). Mineral-associated heavy fraction (HF) had lower C concentrations but comprised the majority of soil mass. Overall, HF represented 40–70% of the C for granites, nephelinite and dry gabbro soils and >80% in



other soils (Table 3). Differences between the sum of LF and HF and bulk C can be either in the mass of fine roots (picked out of the LF-fraction), or material that dissolves in the sodium polytungstate solution.

LF $\delta^{13}$C ranged from -24‰ to -14.5‰ (Table 3), reflecting a mixture of C3 and C4 vegetation sources (Scholes et al. 2003). We found no relationship between LF $\delta^{13}$C or HF $\delta^{13}$C with rainfall, but mafic soils were consistently more enriched than

5 felsic soils (Figure 2a). Radiocarbon signatures of LF (that includes char as well as plant fragments) varied from values close to those measured in annual grasses in 2010 (+35‰ in $\Delta^{14}$C) up to +145‰ (Figure 2b). For surface horizons, the one-pool model yielded two possible turnover times for most of the LF $\Delta^{14}$C. Assuming the shorter of the two for non-mafic soil A horizons (normally 0–2 cm), yielded TT's from <1 to 8 years, while assuming the longer TT yielded 45–185 years (Table 3). For the black basalt and dry gabbro soil A horizons, only longer LF TT's (125–185 yrs) were consistent with observed $\Delta$

$^{14}$C signatures. TT of both LF and HF fractions increased with depth. Fine roots picked from LF in the red basalt soil had radiocarbon signatures equivalent to TT <1 year regardless of depth (Suppl. Table 1), and $\delta^{13}$C signatures of -12 to -16‰.

The $\delta^{13}$C of HF averaged ~ 3–4 ‰ more enriched than $\delta^{13}$C of LF from the same soil (Figure 2a). Radiocarbon signatures in mafic soil HF were generally much more depleted in $^{14}$C than LF from the same horizon (Figure 2b). Felsic soils tended to have higher $^{14}$C values in HF than mafic soils, though this was less the case for LF fractions.

### 3.3 Depth Profiles

Soil depths increased with rainfall from north to south. In all soils, C and $^{14}$C concentration decreased with depth. For C concentration and isotopes, differences in lithology and hence mineralogy were more important controls than differences in

rainfall (Figure 3 and Supplemental Table 2). Soils developed on nephelinite had the highest C concentrations, while felsic soils had the lowest (Figure 3).

Depth profiles of $\delta^{13}$C followed two general patterns. Granitic, gabbro and nephelinite soils had large (2–6‰) increases in $\delta^{13}$C between the surface and ~10–30 cm depth but declined again with depth (Figure 3). Red and black basalt soils increased only ~2‰ in $\delta^{13}$C with depth and then stayed constant. Radiocarbon declined with depth in all soils, but in wetter

sites (>550 mm annual rainfall) shifted towards higher $^{14}$C values at the very bottom of the profile (BC or C horizons; Suppl. Table 1). The $^{14}$C signatures of organic C in the red and black basalt soils was lower (<0‰ at all depths, even at the surface) compared to the other soils, and were the most enriched in $\delta^{13}$C at all depths (Figure 3).

The B horizons of the red basalt and the two dry gabbro soils contained pedogenic carbonates at concentrations of up to several percent and radiocarbon ages ranging from ~4500–25000 $^{14}$C years, substantially lower in $^{14}$C compared to organic C

at the same depths (see Suppl. Table 1). The carbonates in B1 horizons were generally found as small fragments (with $^{14}$C ages up to ~4500 y) and likely derived from older (up to 25,000 years in radiocarbon age) and more massive carbonates deeper in the soil.



## 3.4 Clay-associated C

The C concentration of the clay-size fraction ranged 1.0–4.7% across all soils, with the highest %C in clays having the greatest proportion of smectite (red and black basalts; Table 4). The radiocarbon-estimated TT of organic C in the clay-size fraction ranged from 310-1330 years in the A horizon (Table 4). Clay-C TT increased with depth (data from horizons extending >20-cm depth are not shown). The TT estimated for clay-associated C in the top two horizons increased with abundance of smectites in the clay (Figure 4, left).

Assuming bulk C has two components, clay-associated and non-clay associated, we estimated the amount and radiocarbon signature of the non-clay component from mass balance:

$$\%C_{non\text{-}clay} = (\%C_{bulk} - F_{clay} \times \%C_{clay})/ (1 - F_{clay}) \tag{1}$$

$$\Delta^{14}C_{non\text{-}clay} = \Delta^{14}C_{bulk} - F_{clay} \times \Delta^{14}C_{clay})/(1 - F_{clay}) \tag{2}$$

where $F_{clay}$ is the fraction of bulk-C that is found in the clay fraction:

$$F_{clay} = (\%C_{clay} \times \%clay)/(\%C_{bulk} \times 100\%) \tag{3}$$

For the basalts, with mostly smectite, $F_{clay}$ stabilized 40–47% of $C_{bulk}$ in the top 2 cm, increasing to 80–86% deeper in the B horizon (see Suppl. Table 1). For all other soils, the amount of C associated with clay accounted for <30% of bulk C. Other than the high-smectite basalt-derived soils (and deeper B horizons in the gabbros; Suppl. Table 1), most bulk C was stabilized by mechanisms other than sorption to clay surfaces. As estimated from mass balance, $\Delta^{14}C_{non\text{-}clay}$ in the top 18 cm was (with one exception) dominated by 'bomb' $^{14}C$ $\Delta^{14}C > 0$‰); i.e. dominated by C fixed in the last 50 years (Table 4). The estimated TT for non-clay stabilized C ranged from 30-690 years, averaging 190±190 years; Table 4). The clay-associated C for the same samples averaged 1020 ± 460 years.

The fractionation methods applied in this study, based on density and size, overlapped in terms of the isolated soil fractions (Figure 4). For example, much of the C in the clay-sized fraction can be assumed to make up part of the measured HF. Although it is generally assumed that weak (2%) $H_2O_2$ does not oxidize clay-bound organic C, and the %C measured in clays mostly exceeded concentrations of C found in HF or bulk soil, we cannot rule out that some C was removed from this fraction (Figure 4). There is also overlap between the LF and the non-clay associated C fractions. Although again we assume the $H_2O_2$ would have removed much of the LF, the non-clay fraction is estimated by mass balance and thus would include its mass and isotopic signature (Figure 4). The distribution of isotopes and C among the various fractions for one soil (Figure 4) demonstrate these variations and relationships, and show that the biggest differences in radiocarbon are between clay-associated C (oldest) and non-clay or LF fractions (youngest, and very similar).

## 3.5 Mineral-C relationships between profiles

Profile-averaged properties were calculated to emphasize the factors that controlled differences between soils across landscape elements (Table 5). This calculation introduced errors associated with highly uncertain estimates of gravel content




and bulk density (see values in Suppl. Table 1), but the errors introduced in our analysis are identical for the elements being compared (e.g. profile-averaged concentrations) so profiles being compared should share systematic biases (i.e. operator error similar everywhere). The exception is the calculation of total profile C inventories (Table 5) that are included to demonstrate the importance of these factors in understanding landscape-scale C storage and dynamics. For example, the

nephelenite soil had highest C concentrations (averaging 3.8%C over the whole profile) but was estimated to have 80–90% gravel (Supplemental Table 1), so the estimated C inventory (1.1 kg C m$^{-2}$) is not the highest when compared to other soils (which ranged from a low of 0.6 kg C m$^{-2}$ in the dry granite soils to 11.4 kgC m$^{-2}$ in the black basalt soils). Thus it should be remembered that the relationships derived below are for the <2-mm component of soil.

Mass-weighted mean profile %C$_{organic}$ correlated significantly with mineral CEC (i.e. CEC corrected for organic matter

contribution), and Fe(d) (Figure 6 and Suppl. Table 2). Dithionite extractable Fe and non-organic CEC together explained most of the variation in carbon inventory across all soils. At the soil profile scale, we found a significant relationship between the amount of smectite and the mean TT (Figure 6); total clay and smectite each also correlated with $^{14}$C, but not as well as their product (see complete correlation matrix in Supplemental material, Table 2). The only highly significant correlation for $^{13}$C was with $^{14}$C (Figure 6), though less significant correlations were found between $^{13}$C and average clay

content, and pH and CEC (Suppl. Table 2).

# 4 Discussion

Geological, climatic and topographic variation in Kruger National Park give rise to soils of varying mineral compositions.

Different stabilization capacities and timescales associated with these minerals lead to observed patterns in C inventory and TT across the sampled landscape. None of the soil properties we measured showed a significant relationship with mean annual precipitation (Suppl. Table 2), indicating that any influence of climate on C cycling was indirect, through mineral composition and possibly vegetation. Parent rock lithology significantly influenced clay content, smectite content, cation exchange capacity, and bulk C turnover time (Suppl. Table 2).

By sampling across broad gradients in soil forming factors and thus a range of mineralogy, our goal was to determine whether there were simple, scalable relationships between measures of soil mineralogy with C inventory and C turnover. Overall, we find that no single mechanism can explain all the C properties of these soils, in part because of differences in the depth-dependent distribution of controlling variables, but primarily because of different mineral compositions.

Our initial hypothesis was that, in the absence of SRO minerals, the amount of smectitic versus kaolinitic clays would be the

strongest influence on C inventory and C turnover. Soils in the KNP indeed contain few SRO minerals. C associated with smectite clay had TT averaging ~1000 years in horizons within the top 2-15 cm, and the amount of smectite clay correlated (weakly) with the age of clay-C in A horizons (Figure 5a). Using a larger subset of data (since we measured 14C and clay



mineralogy in only a subset of samples), we also found correlations (Figure 5b) between the fraction of clay-associated C and the TT of bulk C in A horizons.  At the scale of the whole profile, the amount of smectite clay correlated significantly with mean TT of C (Figure 6).  Hence, we conclude that C associated with smectite clay surfaces is strongly stabilized, and that the amount of smectite clay exerts a control on the overall age of organic C in soil profiles of the KNP. We did not

measure radiocarbon directly in kaolin-rich clays from our sites.  However, soils at the toeslope of the granitic catena with HF C radiocarbon signatures > +50‰ still had 39–49% clay (~23–26% of which was smectite and 53–57% kaolins; Table 2). Hence it is not the amount of clay in the soil, but the amount of the clay that is smectite that is key to long-term C stabilization in Kruger soils.

These results are in accord with findings by Wattel-Koekkoek and Buurman (2004) that C stabilized on smectite in surface

horizons has turnover times of 600-1400 years in soils from Africa and South America. Wattel-Koekkoek et al. (2003) also showed that ancient C associated with smectite tends to be more aromatic, which suggests smectites provide a long-term store for fire-derived C. The aging of LF C with depth in fire-prone soils was shown to be related to the presence of char in soils from other fire-adapted ecosystems (Koarashi et al. 2013;  Heckmann et al. 2009); where we analysed this in red basalt soils we found increased age of LF C with depth as well (Suppl. Table 1). Though we did not measure the chemistry of LF C

in this study, the presence of charred materials provides one possible reason for the low TT of LF C, particularly in the red and black basalts that had highest smectite concentrations. The basaltic terrain in KNP is characterised by high grass biomass and frequent fires (Govender et al. 2006).

Silicate clays provide just one way to stabilize C in KNP soil. For soils other than those developed on basalts and the deep horizons in gabbro, <25% of the C was stabilized by silicate clay minerals (Table 4). Given the very strong relationship

between dithionite-extractable Fe and %C across our soils (Figure 6), it is reasonable to propose crystalline Fe and Al (oxyhydr)oxides as dominating other, non-clay mineral stabilization mechanisms for organic C.   This fraction, which is calculated as the difference between bulk and clay fractions, also contains up to 20% of C derived from LF and very fine roots.   We expect crystalline Fe and Al (oxyhydr)oxides to be present as coatings on larger sand- and silt-sized materials, or as the cement holding together stable aggregates.  The TT of C associated with the non-clay fraction averaged 190±190

25   years in surface soil horizons (<20-cm depth; Table 4). We thus expect millennially-aged C associated with smectite clays to remain relatively insensitive to future changes in climate and land-use, while the decadal-centennial cycling C associated with sesquioxides and non-smectite clay like kaolinite will respond faster.  For example, with one exception, all of the non-clay stabilized C has incorporated a significant amount of bomb radiocarbon in the past ~50 years (Table 4).

Averaged for the whole soil profile, clay content was not the best predictor of the amount ($r^2$ =0.50, p=0.02) or TT ($r^2$ =0.59,

p=0.03) of soil C, though this relationship improved when only smectite clay was considered (Figure 6; $r^2$ =0.75, p=0.001). Given the long TT associated with C stabilized by smectite, we conclude that even small addition of millennially-aged, smectite-stabilized C contributes substantially to the TT.  For example, mixing 75% C with a TT of 25 years with 25% C with a TT of 1200 years yields a bulk TT of ~320 years.  Increasing the millenial pool to 35% changes the mean age of bulk C to ~450 years.  Thus, the very slow TT of smectite stabilized C relative to other mechanisms dictates that small changes in



the amount of smectite will significantly influence the observed bulk TT.  The same is not true for C stocks, however, which are not significantly correlated with either clay ($r^2$ =0.50, p=0.08) or smectite clay ($r^2$ =0.45, p=0.12), reflecting that most of the C in these soils is not stabilized on clays.

Somewhat unexpectedly, soils with the highest concentration of SRO minerals, the subhumid gabbro and arid nephelinite had younger C than would be predicted based on expected relationships between SRO minerals and C age in many soils (c.f. Torn et al., 1996; Kramer et al., 2012). SRO minerals are particularly strong sorbers of C because their hydrated nano-crystals create intimate mixtures of mineral and organic material that -  in the absence of drying and rewetting or redox pulses -  tend to remain very stable (Chorover et al., 2004; Thompson et al., 2006b: Buettner et al., 2014). However when SRO minerals are subjected to drying and rewetting or oxidation-reduction pulses, they reorganize into larger, more well-order crystalline compounds by ejecting C and water from the interior of their lattice structure (Ziegler et al., 2003; Thompson et al., 2006a). The wet gabbro and nephelinite soils had younger C, and only 9-17% of the C was associated with the clay fraction, even though they have >5% SRO mineral concentrations. The young C suggests that the SRO mineral phase does not have a long residence time in these soils but likely is a relatively transitory phase that forms as primary minerals in the gravel and cobble fraction of the soil weather and ripen to kaolin mineral phases quite quickly, with any C that was sorbed into the SRO mineral phase made available for microbial decomposition.  In the same way, redox oscillations under seasonal wet-dry cycles promote crystallinity of Fe and we suggest that the Fe-bearing SRO minerals in these environments are likely short-lived giving way to crystalline Fe forms where C is sorbed to surfaces rather than within the lattice (Chorover et al., 2004). Thus, although the availability of large surface area may promote stabilization of large amounts of C in these soils (e.g. nephelinite in Figure 3), the short residence time of the SRO minerals themselves, combined with the short TT of C sorbed onto 1:1 clays and crystalline Fe (oxyhydr)oxides, could explain the overal younger ages of C in the nephelinitie and wet gabbro soils.  Studies of mineral-C interactions must consider not only the strength of C association with various mineral phases (strong for SRO and smectite, weak for kaolinite and sesquioxides), but also the timescale of mineral stability in the soil profile and pedogenic setting. Where SRO minerals and sesquioxides are stable, the associated C tends to be old, but in soils such as Kruger Park the combination of a relatively short but strong rainy season and a long intervening dry season can lead to relatively rapid mineral transformation and hence rapid C TT.

Factors that vary with soil depth exert controls on both C inventory and TT in KNP soils, as has been reported in many other areas.  These affect the [14]C in all fractions: estimated TT of LF, clay, clay-associated and non-clay C all increase with soil depth. However, the rates at which age increased with depth differed between soils and C fractions. Where C was mostly smectite-stabilized (e.g. basalts), the age offset between clay and non-clay was largest at the surface and smallest at depth. In contrast, when smectite was not the dominant stabilising agent, the offset between clay and non-clay fractions was relatively uniform with depth. While depth-related increases in C TT may be partly controlled by changes in the age of C associated with the millenial C fraction and with changes in the relative amount of C stabilized on smectite versus sesquioxides, much more work is required to understand the stability of the different mineral phases themselves, especially





Fe- and Al- stabilized organic C, and how they interact with transport mechanisms in soil (e.g. Schrumpf et al. 2013; Schrumpf and Kaiser, 2015).

Mineral/lithologic stabilization also exerts control on $^{13}$C variation in KNP soils, in at least two ways. First, the LF $\delta^{13}$C seem to indicate greater C3-derived vegetation inputs to soils with more felsic parent materials, and a predominance of C4 inputs in more mafic soils (Figure 2a). This is consistent with the vegetation patterns on the ground (Scholes et al. 2003), with C4 grasses dominating the basalt soil landscapes. These patterns are largely preserved in the mineral-associated C (Fig. 2a), although HF $\delta^{13}$C is enriched compared to the LF $\delta^{13}$C. At the profile scale, the strongest predictor of $\delta^{13}$C is $\Delta^{14}$C (or TT; Fig. 6; $r^2$ =0.75, p=0.005), followed by pH ($r^2$ =0.65, p=0.016) and clay content ($r^2$ =0.65, p=0.017; (see Suppl. Table 2). However, for the isolated clay fraction, the relationship between $\delta^{13}$C and smectite was weak. Soils with the highest smectite are also those with the greatest C4 vegetation, so it is unclear whether lithologic control on C3 versus C4 plants, or fractionation associated with different mineral stabilization mechanisms, is responsible for the overall trends observed in $\delta^{13}$C. Nonetheless, interpretations of paleo-vegetation from bulk soils must be undertaken with care, as variations in the mechanism of C stabilization across the landscape may affect the $\delta^{13}$C signature as well as vegetation changes. More work is needed to disentangle these relationships at bigger scales encompassing climate and topographic gradients that will also involve changes in mineralogy.

Large parts of the globe contain old soils with low concentrations of SRO minerals (Paton et al. 1995). We found good agreement in the age of C in smectites and those reported by Wattel-Koekek et al. (2003) from soils collected in prior decades at other sites in Africa and South America, indicating that smectite content will provide a useful indicator for the fraction of C stabilized on millennial timescales over a large area. While clay mineralogy is not an easy measurement, the amount of smectite in our soils was broadly predictable from lithology and from more easily measured soil properties such as CEC (corrected for organic contributions). Thus, across a range of landscapes and parent materials, one could predict how much of the C in soils is cycling on faster and slower timescales based on these parameters, while overall C inventory is more related to crystalline Fe and Al (oxy)hydroxides.

# 5 Conclusions

In KNP, variations in C cycling rates among soils differing in parent material, topographic setting and rainfall received are largely explained by mineral concentrations. The age of C associated with clay minerals depends critically on the concentration of smectite in these soils that have few SRO minerals. However, in all soils except basalts, most C is not associated with clay minerals. Our data indicate that heavy fraction C consists of two major components: a relatively 'passive' pool that stabilizes C for millennia on smectite, and a more dynamic pool stabilized for decadal timescales and likely associated with crystalline Fe and Al (oxy)hydroxides that has $^{14}$C signatures similar to those of the LF fraction. Most C in surface horizons, even in basalt soils where clays are >95% smectite, is in this faster-cycling pool. A small but highly




dynamic light fraction pool also occurs across all soils. Increases in age of C with depth in soil profiles may indicate rates of vertical mixing or the time required for repeated sorption/release of C as it moves downwards, or it may reflect changes in the stability of the minerals themselves as a function of soil depth.  While more research will be needed to understand these issues, our results hold great promise for predicting C inventory and TT based on intrinsic timescales of C stabilization

mechanisms.

**Acknowledgements**

This project was funded by grants from the Andrew W. Mellon Foundation to SET and OAC. We thank Ines Hilke and Birgit Frohlich for the CN analysis, Iris Kuhlman, Marco Pohlmann and others at the Max Planck Institute for
Biogeochemistry for technical support. Thanks to South African National Parks for granting permission to work in the Kruger National Park and logistical facilitation, especially Patricia Khoza and Jacob Mlangeni. We thank Xioamei Xu and staff at the W. M. Keck Carbon Cycle Accelerator Mass Spectrometer, University of California at Irvine for radiocarbon data, and Jun Koarashi also helped with running samples at UC Irvine. William Benzel and George Breit provided invaluable assistance with mineralogical analysis. Shaun Levivk supplied the map for Figure 1. We thank Max for thought-
garden space in SB and the following postdocs and graduate students at UCSB for insightful comments on an earlier version of the manuscript: Joseph Blankinship, Yang Lin, Eric Slessarev, Nina Bingham.

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



Table 1. Identifiers, descriptions and locations of the soil profiles used for this study.

| Identifier | Lithology | MAP (mm) | Slope Position | Year | Latitude (easting) | Longitude (northing) | Classification |
|---|---|---|---|---|---|---|---|
| GR-450-C | Granite | 450 | Crest | 2004 | 322713 | 7452153 | Haplocambid |
| GA-450-C | Gabbro | 450 | Crest | 2010 | 321956 | 7449291 | Calciustoll |
| RH-450-C | Rhyolite | 450 | Crest | 2010 | 351375 | 7421676 | Haplocambid |
| NE-450-C | Nephelinite | 450 | Crest | 2010 | 336567 | 7398988 | Ustorthent |
| BB-450-C | Picrite (black basalt) | 450 | Crest | 2009 | 341888 | 7420588 | Calciustert |
| RB-450-C | Basalt (red basalt) | 450 | Crest | 2009 | 344120 | 7421754 | Duritorrand |
| GR-550-C | Granite | 550 | Crest | 2006 | 348678 | 7231971 | Ustorthent |
| GR-550-S | Granite | 550 | Seepline | 2006 | 348755 | 7231990 | Dystrustept |
| GR-550-T | Granite | 550 | Footslope | 2006 | 348831 | 7231986 | Natrusalf |
| MG-550-C | Mixed granite | 550 | Crest | 2010 | 341298 | 7232342 | Ustorthent |
| MG-550-C2 | Mixed granite | 550 | Crest | 2010 | 341298 | 7232342 | Ustorthent |
| GA-550-C | Gabbro | 550 | Crest | 2005 | 333525 | 7230774 | -- |
| GA-740-C1 | Gabbro | 740 | Crest | 2010 | 329124 | 7218015 | Haplotorrert |
| GA-740-C2 | Gabbro | 740 | Crest | 2010 | 329124 | 7218015 | Haplotorrert |
| GR-740-C | Granite | 740 | Crest | 2004 | 0326823 | 7211630 | Dystrustept |





Table 2. Clay mineralogy for selected soils (mostly mafic soils that had higher clay contents). Abbreviations: Q= Quartz, F= Feldspars, C= Calcite, O= Oxides, K= Kaolins, S=Smectites, Ch= Chlorites, M= Micas

| Identifier | Hor | Clay (%) | Q | F | C | O | K | S | Ch | M |
|---|---|---|---|---|---|---|---|---|---|---|
| | | | ------------------% of clay-sized fraction ----------------------- | | | | | | | |
| NE-450-C | A | 30 | 1 | 1 | 0 | 15 | 24 | 47 | 0 | 12 |
| NE-450-C | Bw1 | 40 | 1 | 1 | 0 | 14 | 26 | 48 | 0 | 9 |
| BB-450-C | A1 | 39 | 1 | 0 | 0 | 0 | 6 | 92 | 0 | 0 |
| BB-450-C | Bw1 | 43 | 2 | 0 | 0 | 0 | 5 | 93 | 0 | 0 |
| RB-450-C | A1 | 36 | 1 | 0 | 0 | 0 | 0 | 99 | 0 | 0 |
| RB-450-C | Bk2 | 46 | 2 | 0 | 0 | 0 | 0 | 98 | 0 | 0 |
| GA-450-C* | A | 15 | 2 | 2 | 0 | 7 | 0 | 67 | 0 | 22 |
| GA-450-C* | Bw1 | 25 | 1 | 9 | 6 | 10 | 0 | 43 | 3 | 28 |
| GA-740-C1 | A | 20 | 0 | 1 | 0 | 14 | 10 | 60 | 6 | 9 |
| GA-740-C2 | Bw1 | 25 | 0 | 1 | 0 | 8 | 16 | 68 | 6 | 1 |
| GA-740-C3 | Bw2 | 10 | 0 | 0 | 0 | 14 | 3 | 77 | 3 | 3 |
| GR-550-C† | A | 14 | 0 | 0 | 0 | 0 | 79 | 0 | 21 | 0 |
| GR-550-C† | Bw2 | 17 | 0 | 0 | 0 | 0 | 79 | 1 | 21 | 0 |
| GR-550-S† | A | 6 | 0 | 0 | 0 | 0 | 76 | 17 | 7 | 0 |
| GR-550-S† | Bw2 | 7 | 0 | 0 | 0 | 0 | 65 | 25 | 11 | 0 |
| GR-550-T† | A | 25 | 0 | 0 | 0 | 0 | 57 | 26 | 17 | 0 |
| GR-550-T† | 2Btn2 | 47 | 0 | 0 | 0 | 0 | 53 | 23 | 15 | 10 |

† Data are from Khomo et al. (2011).

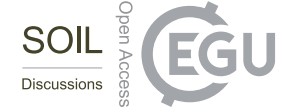

Table 3. Carbon and carbon isotope signatures of heavy (HF, >1.7 g/cc) and light (LF) fraction carbon. Turnover times (TT) were estimated using the one pool model described in the text. When two TT were possible, we show both options for the LD fraction, but only the longer one for the HD fraction. TT were estimated according to the year of sampling (see Table 1), which ranged from 2004-2010.

| Identifier | Depth (cm) | Total C in HF | Low density (LF) | | | | HF (HF) | | | |
|---|---|---|---|---|---|---|---|---|---|---|
| | | | $C_{org}$ % | $\delta^{13}C$ ‰ | $\Delta^{14}C$ ‰ | TT yr | $C_{org}$ % | $\delta^{13}C$ ‰ | $\Delta^{14}C$ ‰ | TT yr |
| GR-450-C | 0-23 | 0.59 | 37.0 | -23.7 | 99.3 | 5, 75 | 0.5 | -19.5 | 30.1 | 195 |
| | 23-45 | 1.06* | 24.6 | -18.9 | 125.6 | 8, 50 | 0.4 | -17.5 | -35.9 | 510 |
| GA-450-C1 | 0-2 | na | 34.2 | -16.8 | 44.1 | 155 | na | -13.6 | 6.2 | 275 |
| | 2-12 | na | 42.3 | -15.0 | 34.4 | 180 | na | -14.5 | 20.9 | 225 |
| RH-450-C | 0-3 | 0.81 | 16.6 | -23.2 | 98.4 | 8, 75 | 0.7 | -18.2 | 72.9 | 55 |
| | 3-15 | 0.83 | 1.9 | -20.0 | 76.7 | 5, 100 | 0.5 | -16.2 | 40.9 | 165 |
| NE-450-C | 0-2 | 0.65 | 31.3 | -20.6 | 88.3 | 8, 85 | 4.2 | -16.7 | 74.4 | 105 |
| | 2-18 | 0.76 | 30.8 | -19.5 | 64.4 | 4, 120 | 2.4 | -14.9 | 1.8 | 300 |
| RB-450-C | 0-4 | 0.82 | 11.6 | -16.6 | 60.0 | 3, 125 | 1.8 | -14.8 | -23.1 | 425 |
| | 4-15 | 0.69 | 10.0 | -16.6 | 30.0 | 195 | 1.4 | -13.3 | -95.2 | 985 |
| | 15-30 | 0.92 | 10.0 | na | na | na | 1.4 | -13.3 | -152 | 1560 |
| | 30-49 | 0.84 | 11.2 | na | -53.0 | 330 | 1.3 | -13.3 | -216 | 2300 |
| BB-450-C | 0-3 | n.d | 10.0 | -15.4 | 33.5 | 185 | 1.5 | -13.7 | -25.4 | 440 |
| | 3-11 | 0.78 | 7.5 | -16.6 | 33.5 | 185 | 1.6 | -12.9 | -65.6 | 735 |
| GR-550-C | 0-15 | 0.70 | 16.0 | -22.2 | 57.7 | 1, 130 | 0.5 | -18.1 | 50.3 | 145 |
| | 15-41 | na | 14.5 | -21.9 | 82.3 | 2, 95 | 0.4 | -17.1 | 59.0 | 130 |
| GR-550-S | 0-2 | 0.72 | 22.5 | -20.8 | 62.4 | 1, 120 | 0.5 | -18.8 | 62.9 | 120 |
| | 2-10 | 1.1* | 21.3 | -20.2 | 82.1 | 4, 95 | 0.4 | -19.1 | 96.1 | 80 |
| GR-550-T | 0-8 | 0.91 | 21.2 | -18.0 | 54.7 | 1, 135 | 0.8 | -16.7 | 58.0 | 130 |
| | 8-15 | 0.69 | 15.2 | -19.9 | 74.4 | 105 | 0.5 | -17.4 | 79.0 | 100 |
| MG-550-C | 0-3 | 0.84 | 36.9 | -22.3 | 71.7 | 5, 125 | 0.9 | -16.6 | 60.2 | 125 |

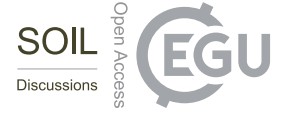

| | | | | | | | | | |
|---|---|---|---|---|---|---|---|---|---|
| | 3-10 | 0.92 | 15.5 | -20.6 | 80.5 | 6, 110 | 0.7 | -15.3 | 54.1 | 135 |
| GA-550-C | 0-9 | na | 29.9 | -28.2 | 98.3 | 5, 85 | 2.8 | -14.4 | 45.8 | 140 |
| | 9-24 | na | 15.2 | -13.2 | 88 | 4, 100 | 2.3 | Na | 38.4 | 175 |
| GA-740-C1 | 0-3 | 0.70 | 34.7 | -16.9 | 72.6 | 1, 100 | 1.6 | -13.4 | 78.2 | 100 |
| | 3-9 | 0.73 | 23.2 | -14.8 | 40.5 | 165 | 1.5 | -11.8 | 6.1 | 275 |
| GA-740-C2 | 0-4 | 0.85 | 35.7 | -16.1 | 67.6 | 4, 135 | 1.5 | -13.1 | 85.2 | 90 |
| | 4-24 | 0.76 | 35.3 | -16.9 | 64.4 | 4, 140 | 1.8 | -12.8 | 60.6 | 125 |
| GR-740-C | 0-8 | 0.41 | 32.7 | -21.3 | 145.8 | 10, 45 | 1.0 | -16.8 | 143.9 | 45 |
| | 8-17 | 0.66 | 18.6 | -17.8 | 95.0 | 4, 80 | 0.3 | -15.1 | 109.4 | 70 |

* Values >1 indicate the magnitude of errors associated with density separations.



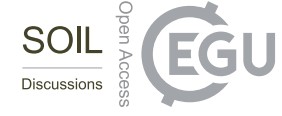

Table 4. Measurements of C and isotopes in the bulk soil and clay-sized fraction analyzed for quantitative mineralogy (Table 2). The fraction of bulk C that is stabilized by clay (F$_{clay}$) and the characteristics of the C not stabilized by clay are calculated using the equations given in the text. Bottom depths >10 cm are B1 horizons; we excluded a few data from sampled depths >20 cm from these comparisons.

| Identifier | Bot. Depth (cm) | Clay (<2-μm size fraction) C$_{org}$ (%) | δ$^{13}$C ‰ | Δ$^{14}$C ‰ | TT (yr) | Bulk Soil Corg (%) | δ$^{13}$C ‰ | Δ$^{14}$C ‰ | TT (yr) | F$_{clay}$ | Carbon not stabilized by clay C$_{org}$ (%) | δ$^{13}$C ‰ | Δ$^{14}$C ‰ | TT (yr) |
|---|---|---|---|---|---|---|---|---|---|---|---|---|---|---|
| NE-450-C | 2 | 2.52 | -16.9 | -1.5 | 310 | 6.04 | -17.8 | 65.0 | 110 | 0.13 | 6.5 | -17.9 | 69 | 110 |
| NE-450-C | 18 | 1.14 | -16.8 | -129.5 | 1330 | 3.04 | -15.4 | 8.0 | 270 | 0.15 | 3.4 | -15.3 | 16 | 240 |
| GA-450-C1 | 2 | 4.69 | -14.8 | -64.6 | 730 | 3.28 | -14.9 | 20.1 | 225 | 0.22 | 2.9 | -14.9 | 58 | 130 |
| GA-450-C1 | 12 | 2.35 | -13.9 | -145.0 | 1485 | 1.90 | -13.9 | -28.8 | 465 | 0.31 | 1.7 | -13.9 | 43 | 160 |
| RB-450-C | 4 | 2.50 | -14.1 | -125.3 | 1280 | 1.94 | -14.9 | -16.0 | 385 | 0.47 | 1.5 | -16.0 | 149 | 30 |
| BB-450-C | 3 | 2.48 | -14.3 | -91.7 | 955 | 2.41 | -13.7 | -25.4 | 440 | 0.40 | 2.4 | -13.4 | 22 | 220 |
| GA-450-C2 | 2 | 1.29 | -16.3 | -4.4 | 320 | 1.62 | -16.6 | 22.0 | 220 | 0.12 | 1.7 | -16.6 | 25 | 210 |
| GA-740-C1 | 4 | 0.96 | -17.6 | -91.7 | 955 | 2.18 | -11.8 | -62.1 | 690 | 0.09 | 2.3 | -11.6 | -61 | 690 |
| GA-740-C2 | 9 | 1.17 | -19.1 | -160.0 | 1650 | 1.76 | -13.6 | 88.1 | 85 | 0.14 | 1.9 | -12.9 | 122 | 55 |
| GA-740-C2 | 24 | 1.10 | -17.7 | -116.1 | 1180 | 2.24 | -13.4 | 70.0 | 110 | 0.10 | 2.4 | -13.0 | 83 | 90 |



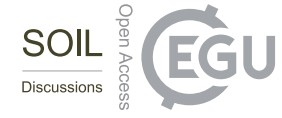

Table 5. Profile-averaged (excluding BC /C horizons) properties for the soils sampled in this study. Averages for C and C isotopes are calculated from bulk values. C inventory (Cinv) is the sum for the profile in kgC m$^{-2}$.

| Identifier | Depth (cm) | Cinv | pH | CEC* | Clay (%) | Smec. | Fe$_d$ | Al$_d$ | Fe$_o$ | Al$_o$ | C$_{org}$ (%) | C/N | δ$^{13}$C (‰) | Δ$^{14}$C (‰) | TT (yr) |
|---|---|---|---|---|---|---|---|---|---|---|---|---|---|---|---|
| RH-450-C | 30 | 1.3 | 6.8 | 5.5 | 1.0 | 10 | 2.7 | 0.2 | 0.1 | 0.1 | 0.66 | 9.9 | -16.2 | 28.0 | 230 |
| GR-450-C | 23 | 0.6 | 6.1 | 7.7 | 6.3 | 46 | 0.6 | 0.1 | 0.0 | 0.0 | 0.78 | 11.9 | -20.2 | 30.4 | 200 |
| NE-450-C | 18 | 1.1 | 6.8 | 61.8 | 38.7 | 48 | 5.4 | 0.5 | 0.3 | 0.2 | 3.85 | 11.8 | -15.9 | 9.9 | 235 |
| BB-450-C | 49 | 11.4 | 7.7 | 44.3 | 42.0 | 98 | 1.7 | 0.2 | 0.2 | 0.2 | 1.56 | 14.2 | -13.5 | -140.4 | 1500 |
| GA-450-C | 34 | 8.6 | 8.3 | 25.7 | 9.9 | 50 | 1.9 | 0.3 | 0.2 | 0.2 | 1.66 | na | -15.0 | -37.0 | 550 |
| RB-450-C | 70 | 8.6 | 7.0 | 50.1 | 46.2 | 93 | 2.5 | 0.3 | 0.1 | 0.1 | 1.53 | 14.8 | -12.3 | -156.4 | 1720 |
| GR-550-C | 62 | 3.5 | 5.4 | 3.2 | 14.8 | 1 | 0.4 | na | 0.1 | na | 0.32 | 21.2 | -15.3 | 12.2 | 430 |
| GR-550-S | 41 | 1.8 | 5.1 | 2.6 | 7.5 | 21 | 0.1 | na | 0.0 | na | 0.23 | 19.6 | -18.6 | 51.8 | 150 |
| GR-550-T | 46 | 3.6 | 7.0 | 29.9 | 42.7 | 24 | 0.2 | na | 0.1 | na | 0.50 | 14.1 | -14.7 | 24.8 | 225 |
| MG-550-C | 38 | 2.6 | 6.9 | 7.8 | 15.0 | 41 | 1.4 | 0.3 | 0.3 | 0.2 | 0.82 | 14.2 | -13.8 | -68.6 | 755 |
| GA-740-C1 | 44 | 7.0 | 7.2 | 37.7 | 17.7 | 69 | 2.6 | 0.2 | 1.1 | 0.4 | 1.53 | 12.2 | -13.9 | -47.5 | 250 |
| GA-740-C2 | 25 | 4.5 | 7.3 | 31.5 | 25.8 | 25 | 2.5 | 0.2 | 2.0 | 0.3 | 1.49 | 11.9 | -13.3 | 26.9 | 150 |
| GR-740-C | 93 | 5.1 | 5.7 | 7.0 | 3.7 | **10** | 0.4 | 0.0 | 0.0 | 0.0 | 0.36 | 14.8 | -18.5 | 88.8 | 240 |

Depth indicates the depth to which the in the profile averages were calculated (we excluded BC /C horizons).

Values in bold for smectite content (Smec.) are assumed based on similar lithology values. We assumed average values for the horizons above and/or below to fill in data for smectite content for depths in a profile where no measurements were available (see Supplemental information).

Subscripts d and o represent dithionite and oxalate extracts for Fe and Al.




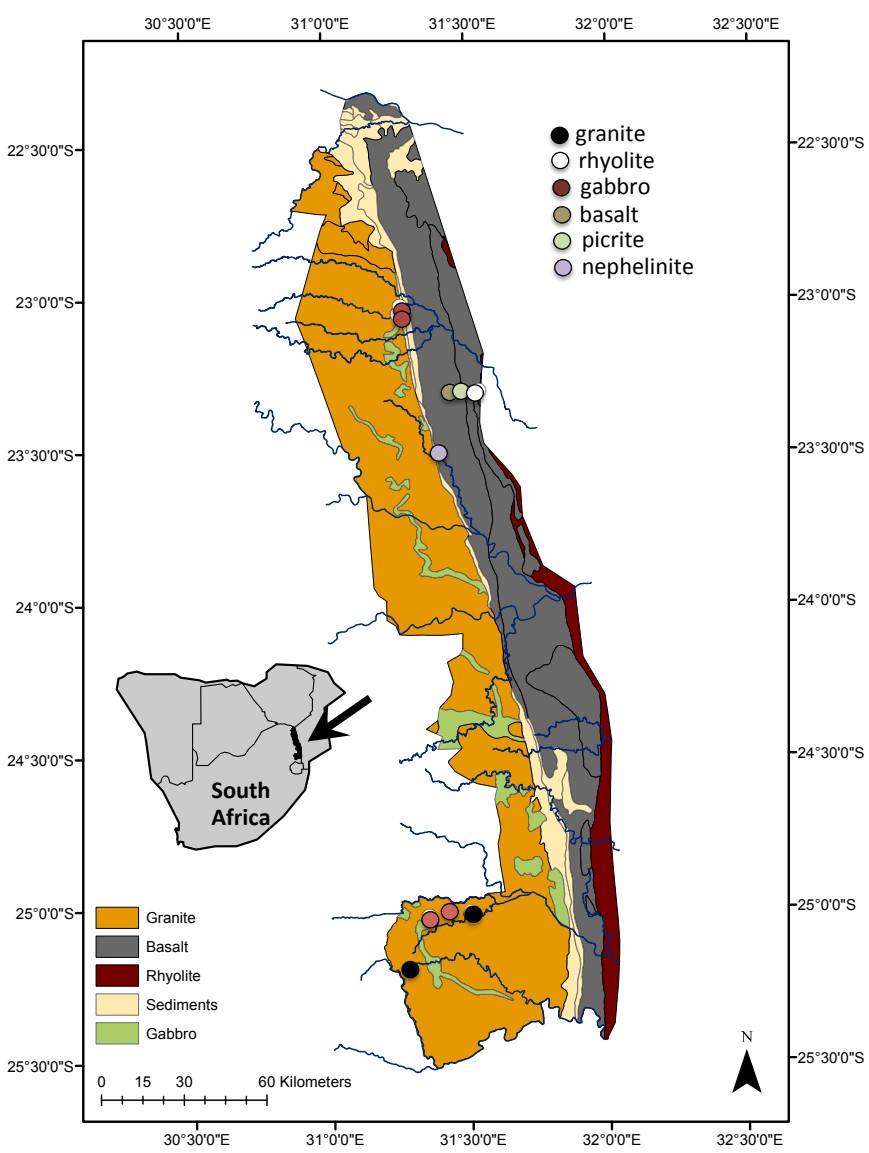

Figure 1. Locations and lithology of soil pits sampled for this study. Rainfall decreases from ~740mm/a in the southern
of the park to ~450 mm/a in the northern end of the park.





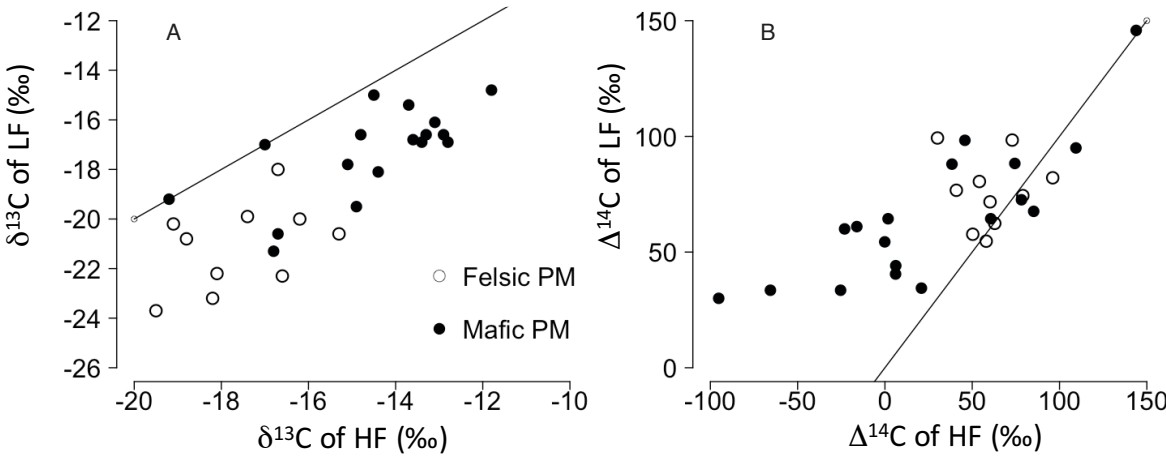

Figure 2. Comparison of $^{13}$C (2a) and $^{14}$C (2b) in low density (LF) and high density (HF) organic C for individual samples from A horizons (see Table 3). Felsic lithologies include granite and rhyolite, mafic includes gabbros and basalts, nephelinite indicates low-Si. The 1:1 correlation line is plotted for reference.



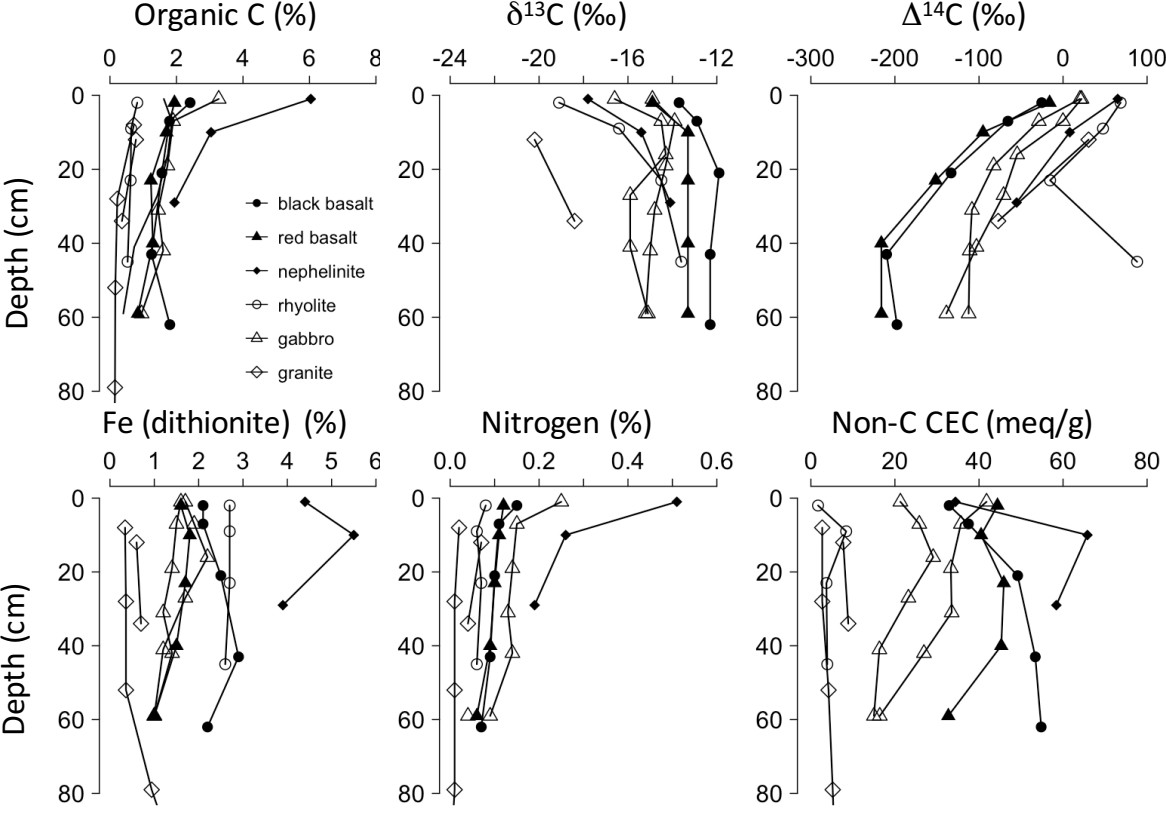

Figure 4.

Figure 3. Depth profiles of bulk C (top-left) [13]C (top-middle), and [14]C (top-right) for selected profiles to compare soils
developed on different lithologies. Also shown are other soil bulk properties, inlcuding Fe(d) (bottom-left), total nitrogen
(bottom middle) and CEC corrected for organic matter contributions (see text; bottom right).



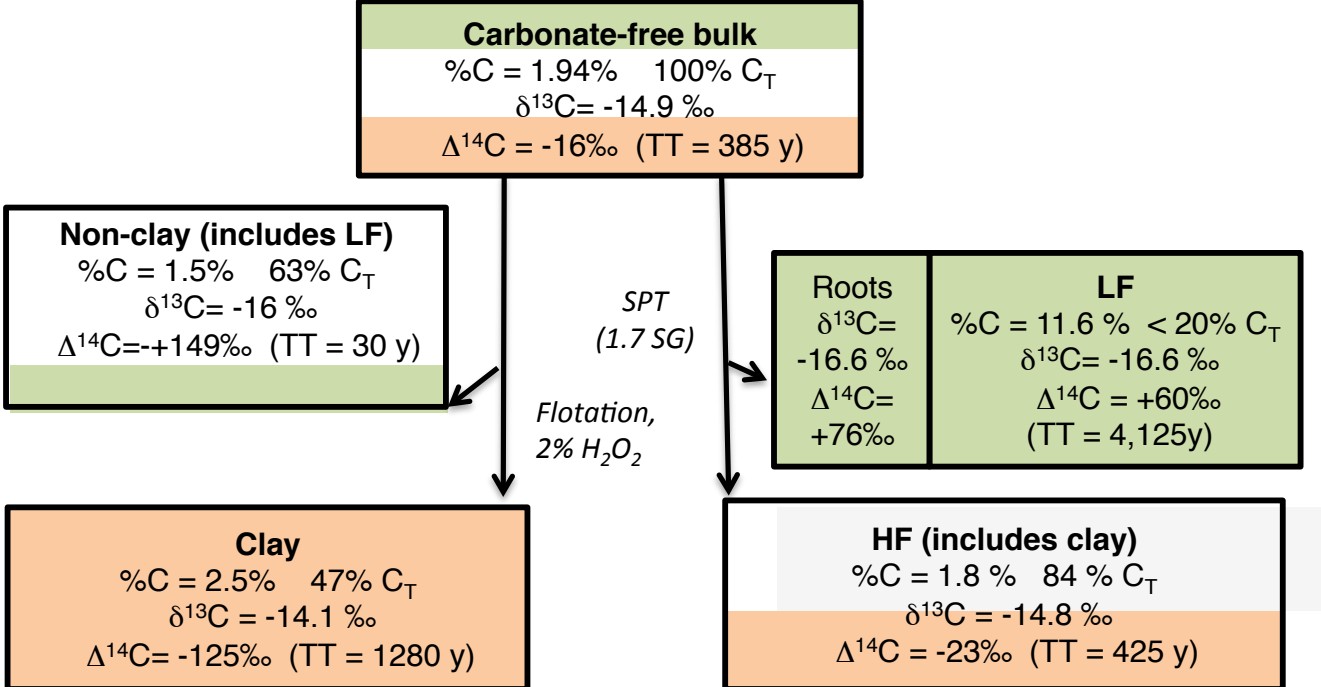

Figure 4. Comparison of C and C isotopes for a single soil sample (0-4 cm depth in RB-540-C (basalt parent material) indicating the inter-relationships among the different process-defined organic C fractions. For each fraction we indicate the percent of total C ($C_T$) it contains; fractions for LF and HF do not add to 100% because of contributions from roots picked from the LF fraction and C that dissolves in the polytungstate solution and is not recovered. Colors indicate overlaps between C among different fractions. For example, clay-associated C (orange) makes up part of the HF-C, and low density C (light green) makes up part of the non-clay fraction. The HF consists of some non-clay fraction (white, indicating the portion not made up of LF).



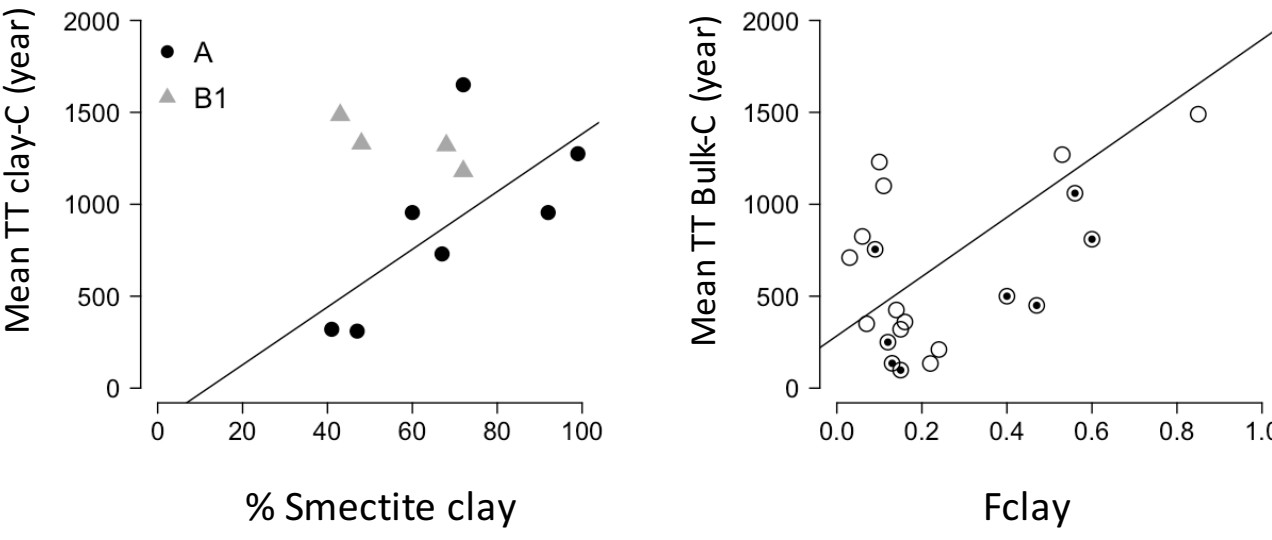

Figure 5. Left - Mean Turnover time (TT) of C in the clay-sized fraction separated from bulk soil. Mean TTof clay-associated organic C increases with depth and with the percent of smectite minerals identified in the clay. Linear relationship for A horizon points (n=7) has R-squared =0.4, p=0.07). Right - Mean TT of bulk C compared to the fraction of bulk C that is associated with C in the clay fraction ($F_{clay}$). This plot has more points because we analysed only a subset of samples for clay mineralogy and clay C isotopes. In these soils with predominantly smectitic clays (Table 2), the mean TT of bulk organic C correlates significantly with clay-associated organic C (linear regression R-squared = 0.45, p = 0.005).


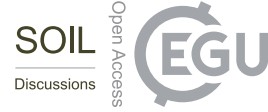

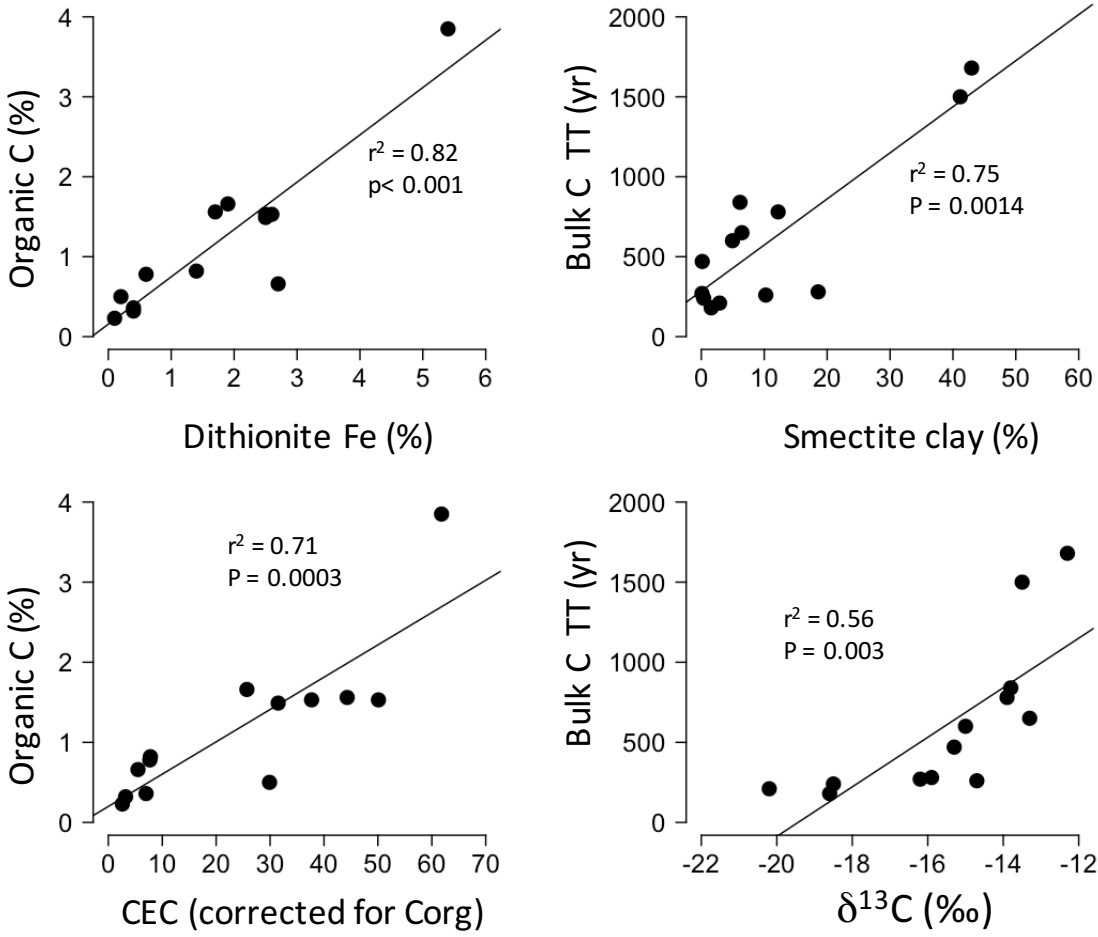

Figure 6. Across all the studied soils, the mean %C in the soil profile was best predicted by Fe(d) (top left), and cation exchange capacity (bottom left). The best predictors for profile-averaged C turnover times was the amount of smectite clay (top right). There is a profile-averaged relationships between $\delta^{13}C$ and the turnover time of organic C (bottom right), and therefore also the amount of smectite clay. Correlation matrices for other variables in Table 5 are given as Supplemental Table 2.