# Peer review of "Timescales of C turnover in soils with mixed crystalline mineralogies"

_SOIL, 2016_

## Referee Comment (RC1) · Anonymous Referee #1 · 11 Jun 2016

General comments

The idea of the present work was to test the organic carbon accumulation and turnover as related to crystalline mineral phases. To do so, soils low in short range order (SRO) mineral phases were studied. The authors report differential effects of certain mineral phases on accumulation and turnover.

I am not convinced that their assumptions are valid. Major misconceptions as well as serious methodological flaws question the entire study. I therefore cannot recommend publication.

Major concerns:

Throughout the manuscript there is a tendency to mix up "accumulation" and "stabili-sation" of organic matter. These are no synonyms. Please try to be specific.

There is also an overall tendency of imprecise expressions and mixing up of concepts. The result is a partly confused text hard to read and understand.

The definition of SRO phases used by authors is rather vague and changes throughout the text. Sometimes it is based in oxalate-extractable Fe and Al, sometimes it seems also the dithionite-citrate-bicarbonate (DCB)-extractable Fe and Al is included.

Analyses of the clay fraction where carried out after treatment with H2O2. This will remove part of the light fraction-type material but also part of the mineral-associated material. Since the exact share of mineral-associated organic carbon removed by the H2O2 may vary, all data organic carbon data obtained on the clay fraction are bi-ased, thus, need to be dropped. Since much of the conclusions are based on the clay fraction-related data this questions the overall meaning of the work. Justifying the analyses of carbon in H2O2-treated material by claiming that some carbon survived the treatment is not valid.

Also, analysis of the mineral assemblage of the clay fractions is based entirely on X-ray diffraction with Cu K$\alpha$ radiation. Thus, there is no proper estimate of possible contents of SRO phases and oxides of the clay fractions. Several sections of the manuscripts left me under the impression the authors are not aware that the clay fraction is not composed entirely of clay minerals but also contains other phases capable to interact with organic matter.

The density separation used is also rather strange. The cut-off density of 1.7 g/cm3 is not justified, at last not by the reference given. In addition, the sonication energy used was little, thus, total dispersion of samples with stable aggregates has to be questioned. Especially, the more oxide and kaolinite-rich samples may not disperse completely, thus, the heavy fraction likely still may contains light fraction-type material. In turn, undispersed aggregates may still contain enough light material to make them float.
The rather variable and often very small contents of organic carbon in the light fractions point at major problems with the separation. In summary, the density separation has been carried out in a hardly reproducible manner.

Minor concerns:

Title: As written, the title suggests a study of only local importance. The authors may consider changing it into "Timescales of C turnover in soils with mixed crystalline mineralogies".

p.1, l. 17-18: The conclusion that the enrichment of OC in the clay fraction is due to stabilisation by clays is either trivial (in case of the authors refer to clay as size fraction) or disputable (in case of referring to clay minerals; see above and below).

p. 1, l. 23: What made the authors believe that crystalline Al oxyhydroxides contributed to the accumulation of faster turning over fraction? Is that based on DCB-extractable Al? If yes, please note that the extractant is not capable of extracting much Al from crystalline Al phases such as gibbsite.

p. 1, l. 26: What does SRO refer to? Seemingly it refers to the observed relationship between DCB-extractable Fe and organic carbon. Dithionite-citrate-bicarbonate-extractable Fe includes Fe from crystalline Fe oxides, such as goethite and haematite.

p.1, l. 29-30 (and elsewhere in the manuscript): Expressions such as "crystalline Fe" are nonsense and misleading. Iron is an element. Please refer to the correct mineral phase, e.g., Fe oxides.

p. 2, l. 3-5: Some references given do not refer to physical protection or recalcitrance. Please be re-consider.

p. 2, l. 7-16: Somehow, I am under the impression the authors have some problems with mechanisms involved in organic-mineral interactions. For example, what is "dehydration bonding"? Also, the choice of references is a bit strange. For instance, two references are on methods for estimating mineral phases but do not address binding

mechanisms. Did Masiello et al. (2004) really refer to crystalline Al and Fe sequioxides?

p. 2, l. 10: Replace "metallic" by "metal".

p. 3, l. 13-16: Note, the studies of Wattel-Koekoek et al. (2003, 2004) include no proper estimates of possible contents of oxides, thus, do not allow for distinguishing of effects by clay minerals and other phases, including SRO phases.

p. 3, l. 17-23: I suggest adding proper research questions and/or hypotheses.

p. 3, l. 27-28: What is "soil residence time"? Please explain.

p. 4, l. 24-25: Please give more information on the determination of the clay contents. Did the procedure involve pre-treatments such as destruction of organic matter and oxides?

p. 4, l. 32-33: The location of the CN analyser is probably of no importance. Omit.

p. 5, l. 3: As stated, it seems only the characterisation of clay minerals was attempted.

p. 5, l. 14-16: What was the idea behind adding Al oxide to the clay fractions before X-ray diffractometry? Why not quartz? The addition of Al oxide limits estimation of Al oxide phases. Was the quantification of X-ray diffraction data supplemented by data on the elemental composition? What software was used?

p. 5, l. 24: Why a density-off of 1.7 g/cm3 was used? The reference given refers to 1.6 g/cm3.

p. 5, l. 25: What was the reason of using varying amounts of sample for the density separation? Why no standard protocol was used? Did the authors attempt estimating the carbon recovery?

p. 5, l. 26: What is meant with "ultrasonicated at 60 J m-1 for 2.5. min"? Is 60 J/ml the total energy input? Did the authors control for proper dispersion, i.e., disaggregation?

And if yes, how this was done?

p. 5, l. 29-30: What is the idea behind removing visible roots from the light fraction? Actually, dead roots are part of the debris material that makes up the light fraction. Why the authors did not remove living roots from fresh samples?

p. 6, l. 1-4: Due to the bias in the clay and density separation I do not think the data set to be solid and comprehensive. I am wondering why the authors did not try a more logic approach, such as first separating all light material, then, separating the clay fraction from the heavy fraction.

p. 6, l 13-14: Is acidification to pH 6 really enough remove all carbonate, including that bonded to other mineral phases? I doubt.

p. 7, l. 13: No, DCB does not extract only crystalline Fe oxide phases but all Fe oxide phases, including ferrihydrite. In turn, oxalate extracts only the poorly crystalline portion of DCB-extractable Fe.

p. 7, l. 14: Note, DCB-extractable Al does not represent Al oxide phases.

p. 7, l. 18: Is there are reason why the clay contents increased with depth?

p. 8, l. 27-32: Is it correct, the soils containing pedogenic carbonates are those rich in smectite? So, could it be that their radiocarbon signature was affected by carbonate? By the way, the picrite (back basalt)-derived soil is classified as Calciustert but not listed as containing carbonate? Isn't that illogic?

p. 9, l. 8-9. Due to the $H_2O_2$ treatment of the clay fraction, I doubt that proper estimates on non-clay carbon are possible.

p. 10, l. 9-15: These correlations need to be re-considered bearing in mind that the clay fraction contains most of the oxides.

p. 11, l. 9-17: The discussion here is rather speculative since composition of organic matter was not addressed.

p. 11, l. 18-29: Here, it becomes clear the authors partly equal clay-sized particles and clay minerals. They relate the estimated non-clay fraction carbon to "other, non clay mineral stabilization mechanisms". This is simply wrong, since the clay fraction holds also most if not all oxide phases (even the Cu K$\alpha$ X-ray diffraction, despite of being rather insensitive to Fe oxides, indicated their presence). Also, I have problems with the authors' logic. The organic carbon of all study soils relates well to the DCB-extractable Fe. Seemingly, the clay mineral type does not matter much to the accumulation of organic carbon. So, there is no reason for all the clay mineral discussion. I am also wondering, why the authors did not comment on the rather small contents of organic carbon in the smectite-rich soils. There is much writing on organic matter stabilisation by smectites. The results presented, however, suggest that smectites are rather poor organic carbon accumulators.

By the way, why do the authors consider "crystalline Fe and Al (oxyhydr)oxides" as controlling carbon accumulation? The close relationship is for DCB-extractable Fe and not for any Al. Also remember, DCB extracts all non-crystalline Fe oxides (see above).

Assuming that the relationship between organic carbon and Fe oxides is also valid for the smectite-rich soils, i.e., Fe oxides do the accumulation; it is absurd to assume that the smectites make the organic carbon turning over slowly. As already pointed out, I assume an issue with incomplete removal of carbonates to be the reason of the radiocarbon signature of the smectite-rich soils.

Tables and Figures (general): Please give proper dimensions. Percentages of smectite, organic carbon, smectite likely refer to weight-%. Consider giving instead g/kg. What is the dimension of the cation exchange capacity (CEC)?

None of the tables and figures is truly self-explanatory.

Table 2: I am wondering if the "oxides" included also Al phases. The little clay content of the non-mafic soils is no good reason for not analysing the composition of their clay fractions.

[Figure]

Table 3: The rather low organic carbon contents of some of the light fractions support my concerns on issues with the density separation. The light fractions with little carbon may contain a good portion of mineral-bound (older) carbon, thus, suggesting a slow turnover.

Figure 5: Giving an enrichment factor for clay fractions treated with $H_2O_2$ is not valid.

---

## Referee Comment (RC2) · Anonymous Referee #2 · 15 Jun 2016

GENERAL COMMENTS

Summary: This manuscript investigates the controls on carbon turnover and C inventory across soils ranging in mineralogy in Kruger National Park, South Africa. The authors sampled soil across a variety of parent materials resulting in differing mineralogical characteristics. Older soils with low contents of short-range order mineral phases were chosen to specifically test the impact of phyllosilicate clays on C turnover time and stocks. To elucidate mineral protection mechanisms, particle size and density fractionation was used in combination with 13C and 14C measurements. The main finding is that the content of high surface area, 2:1 phyllosilicates (i.e., smectite) is a better predictor for C turnover times in these soils than clay content (i.e., < 2 $\mu$m size

fraction). They conclude that analysis of phyllosilicate clay composition reveals clearer insights into C stabilization mechanisms than clay content alone.

Strengths: This manuscript provides a unique dataset in that it is one of very few that measured radiocarbon on different soil fractions (particle size and density) of the same soils. This approach puts the authors in a position to examine the usefulness of individual fractionation approaches for different soil system, and provides interesting insights into the mineral protection mechanisms responsible for C storage in low-SRO, phyllosilicate clay dominated soil systems. This manuscript will be of broader interest to the SOIL readership and I support its publication.

Weaknesses: 1) The authors don't draw a clear line between two definitions for 'clay', i.e. clay as a particle size category and clay as phyllosilicates. It would be helpful to be consistent with this terminology throughout the manuscript

2) The manuscript could use some editing. The introduction is not very concise and could state the research question more clearly. There are typos, somewhat confusing sentence structures and word choices, and mislabeled figures throughout (see specific comments below).

SPECIFIC COMMENTS

Abstract

P1L18: What does 'stabilized' mean in this context? Adsorbed?

P1L19-20: This is a speculative argument informed by the data presented here and should be framed as such. It is a valid point to make, but it should not be framed as a result.

P1L21-24: This argument is confusing to me. The authors state that the fraction that is not clay (> 2 $\mu$m) has much shorter turnover time than the clay fraction. They then attribute the shorter TT in the > 2 $\mu$m fraction to weaker bonding of C to Fe and Al oxyhydroxides and kaolinite as well as the presence of more modern plant inputs (light

fraction). The part that's confusing to me here and throughout the manuscript is the fact that I would consider Fe/Al oxyhydroxides and kaolinite as clays. They could be clays either because of their small size (oxyhydroxides) or because they are phyllosilicates (kaolinite). I understand the authors' point in the discussion that some of these minerals may have stuck to larger grains and so were removed with the $> 2~\mu$m fraction. But I strongly suggest making sure you clearly separate when you talk about clays as size separates and when 'clay' means phyllosilicate.

L24: HF and LF are not defined

Introduction

P2L8: What is "dehydration bonding"? Do you mean ligand exchange?

P2L13-15: Poch et al. seems an odd reference here. It's work on clays on Mars.

P2L15-16: Masiello et al found correlations between pyrophosphate extractable Fe and Al and turnover time. I thought that pyrophosphate extracts poorly or amorphous Fe and Al phases, not crystalline sesquioxides as the authors state here.

P2L17-24: This argument seems very convoluted to me. Isn't the argument that organics on different mineral phases may exchange at different rates and therefore have different turnover times. Fractionation techniques average across a number of these interactions, and so there is a need to look into mineral composition of these fractions more closely to gain insights into what mineral phases provides the most protection (and thus the longest turnover time)? I see where the authors are headed with this argument, but this paragraph could be more concise.

P2L28: What is 'older C storage'. Choice of words (c.o.w.) is odd.

P2L32: Lawrence et al provide a large dataset, but it's not true they 'virtually' measured 'all' reactive components in the system.

P3L2-4: These two sentences makes little sense to me. I think it would help to frame

the argument in terms of C stocks and turnover times, rather than 'storage' and 'stability'. That applies to the remainder of the paragraph.

P3L8: 'Trade-offs" of what?

P3L9-10: I agree with this statement, but that is not a research question.

P3L10: Having this very broad goal stated here seems a bit misplaced. I think the authors should state the specific research question here.

Methods

P3L28: 105 yrs? I might be wrong, but shouldn't it be longer?

P4L31: Typo, should be Plasma

Results

P7L19-20: where does it go up to 26%. The values in Table 2 range from 0-15%?

Table 2 was also published previously, better in the SI?

Table 3: I would include total C in HF (mass of C). There are also a few typos in the table header.

P9L7: I think the authors are referring to Fig. 5. The authors should check what figures are referred to in the following. It didn't always seem to match up.

P9L21-29: Why does every sentence refer to Fig. 4? I'm not sure what is said here refers to anything I can see in Fig. 4.

P9L23ff. On the issue of whether or not H2O2 oxidizes mineral bound OM. I think it does, and it's a mute point to argue it doesn't. But the authors could at least find some refs to support the claim that it has minor impacts.

Discussion

P10L27: c.o.w. What are 'C properties'?

P12L6: c.o.w. 'sorbers' is not a word. It's 'sorbents'.

---

## Author Response (AR1)

**Review Discussion**

The comments of both reviewers indicate that our manuscript was flawed due to poor definition and inconsistent use of a number of terms in the paper. For example, we clearly were not consistent or even always correct in our use of the 'clay' as a particle-size designator on the one hand and as a descriptor of crystalline layer silicates on the other. Given the topic of the paper the confusion created by our admittedly sloppy use of terms created understandable frustration for the reviewers. We recognize that a substantial revision of the manuscript is required before the validity of our interpretations can be fairly judged.

Reviewer 1 pointed out numerous instances of confusing terminology and further suggests that our confusion led to misinterpretations of the results. Reviewer 2 echoed many of those concerns but was a bit more open to following our line of argument even when our terms were imprecise. An example might be our use of the phrase "crystalline Fe and Al" which reviewer 1 rightly points out makes no sense. What we should have said was "DCB extracted Fe and Al oxide or hydroxide compounds from which oxalate extracted Fe and Al oxide or hydroxide compounds had been subtracted." Obviously the latter is too cumbersome but the former was too truncated. One can probably get the sense of what we meant from the former. However, such a lack of precision is inappropriate and annoying in a scientific paper.

We hope the editor will allow us to revise our manuscript. Among the things we will change will be to give clear definitions of all of the various terms used in the manuscript, as given below.

One of the major problems we have is that there is no universally agreed upon method to quantify the mineral components in soil. We quantified the crystalline aluminosilicates such as kaolinite and smectite, but also Fe oxihydroxide minerals like hematite and goethite using XRD on clay-sized material isolated from the bulk soil. Separately, we quantified the Fe oxihydroxides and SRO minerals using standard Dithionite citrate and oxalate extractions followed by measurement of Fe in the dithionite citrate extract and Fe and Al in the oxalate extract. A major problem with our originally submitted manuscript was that we did not clearly state in all cases which of the methods was used when we discussed results.

Here we define some of the "mineral" terms we use in this ms and note that we will use these definitions consistently in the revision:

Clay (<2-μm size fraction)

Clay$_{XRD}$ (<2-μm size fraction used for XRD analysis of minerals, treated with 2% $H_2O_2$)

Clay Minerals (all the identifiable minerals that reside in the <2-μm fraction – this includes layer silicate minerals, Fe oxihydroxides, Al oxide, carbonates)

Layer Silicate Minerals (crystalline layered aluminosilicate minerals such as kaolinite and smectite)

Fe oxihydroxides (Fe compounds that can be dissolved using a standard dithionite citrate extraction but which are not dissolved by a standard oxalate extraction – these compounds are assumed to be pedogenic Fe molecules although it is possible that some geogenic compounds are also dissolved by the dithionite extraction). This fraction also includes coatings on minerals with size >2 μm but <2 mm.

XRD-measured Fe oxihydroxides in the clay fraction. These are the Fe oxihydroxides measured by XRD in the <2-μm fraction, as goethite, hematite, goethite, magnetite, maghemite and ilmenite. We normally will not refer to this fraction, as the standard dithionite citrate and oxalate extractions were performed on more soils.

SRO minerals (aluminosilicate or Fe oxihydroxides that are minimally polymerized and tend to be linked to organic compounds or water – effectively these compounds are extracted using a standard oxalate extraction and quantified by measurement of Fe and Al in solution after extraction). In SRO minerals the crystallites are so small that they do not provide a coherent XRD signal

Al oxide or gibbsite is another component of clay minerals but one that we are not explicitly quantifying in this analysis – as noted below we used Al as an internal standard for the XRD quantification of the Layer Silicate Minerals.

A second issue was the nomenclature used to define the different organic matter fractions that were measured for C and $^{14}$C content.

We define our use of organic matter as follows:

Free Light Fraction (fLF) carbon: The organic carbon in material that floats in a solution with density 1.7 g cm$^{-3}$. (Questions by reviewer 2 about details of the density fractionation procedures are given below, this would be considered "free" light fraction based on the weakness of the sonication).

Root-free fLF carbon: The organic carbon remaining once visible roots have been picked out of the light fraction (please see Castanha et al. reference now added to the manuscript, and other comments below). We chose to pick out visible roots because the removal of roots earlier in sample preparation is highly variable.

Heavy Fraction (HF) carbon: The organic carbon in material that sinks in a solution with density 1.7 g cm$^{-3}$.

Carbon strongly associated with XRD-clay. The carbon in the same clay fraction measured by XRD for mineralogy. This is 'strongly bound' because the material measured for XRD was treated with hydrogen peroxide and presumably only material that has a strong association with minerals or aggregates survives this

treatment.   Carbon strongly bounded to clay is a subset of the Heavy Fraction carbon (which can include C associated with Fe oxihydroxides coating sand grains)

Carbon not strongly associated with XRD-clay (Carbon in the 'non-clay-sized' fraction). This is determined by mass balance between the Carbon strongly bound to XRD-clay and the Bulk soil C.  It includes a heterogeneous mix of materials, from very fine roots to C associated with Fe oxihydroxides coating sand grains.

**Research Design**

We know from a couple of decades of work that SRO minerals store a lot of carbon and hold that carbon for long periods of time. As noted by Reviewer 2 there has been far less work done evaluating carbon storage in soils where SRO minerals make up a very small proportion of the clay mineral fraction of the soil. Our goal in this paper was to evaluate carbon storage in soils with low concentrations of SRO minerals where we would expect the heavy carbon fraction to be associated with other clay mineral components.

We agree with both reviewers that overlaps among the different mineral and C fractions we analyzed can be confusing, and that a more streamlined procedure is advisable in the future.  However, we also think that in reporting the data we have, we are able to draw robust conclusions about the role of smectite clays influencing the age of carbon in soils with few SRO minerals.

Detailed responses to each of the reviewers are given below.  To ease reading, we have put our responses in red below  the referee's comment.

**Anonymous Referee #1**

General comments
The idea of the present work was to test the organic carbon accumulation and turnover as related to crystalline mineral phases. To do so, soils low in short range order (SRO) mineral phases were studied. The authors report differential effects of certain mineral phases on accumulation and turnover. I am not convinced that their assumptions are valid. Major misconceptions as well as serious methodological flaws question the entire study. I therefore cannot recommend publication.

We understand the reviewer's frustration with the originally submitted manuscript and hope to convince them that the major errors were in communication rather than interpretation.   We thank the reviewer for their patience in giving such a thorough review.

Major concerns:
Throughout the manuscript there is a tendency to mix up "accumulation" and "stabilisation" of organic matter. These are no synonyms. Please try to be specific. There is also an overall tendency of imprecise expressions and mixing up of concepts. The result is a partly confused text hard to read and understand. The definition of SRO phases used by

authors is rather vague and changes throughout the text. Sometimes it is based in oxalate-extractable Fe and Al, sometimes it seems also the dithionite-citrate-bicarbonate (DCB)-extractable Fe and Al is included.

The text has been extensively revised with these concerns in mind. We hope we are now providing clear definitions and sticking to our terminology.

Analyses of the clay fraction where carried out after treatment with H2O2. This will remove part of the light fraction-type material but also part of the mineral-associated material. Since the exact share of mineral-associated organic carbon removed by the H2O2 may vary, all data organic carbon data obtained on the clay fraction are biased, thus, need to be dropped. Since much of the conclusions are based on the clay fraction-related data this questions the overall meaning of the work. Justifying the analyses of carbon in H2O2-treated material by claiming that some carbon survived the treatment is not valid.

We agree that the $H_2O_2$ treatment removes organic C from the clay fraction. However, our goal was to measure the radiocarbon in C that was in exactly the same fraction as was measured for mineralogy using XRD. As long as the $H_2O_2$ treatment was performed consistently, we do not agree that biases associated with differential efficiencies of removal of organic matter by $H_2O_2$ (e.g. surface Fe oxihydroxide coatings versus layered silicate minerals) make the measurements meaningless. In a revision, we will take care to point out that there **are** potential biases, but feel that the comparison of the minerals and age of carbon strongly associated with XRD-clay from the same fraction is useful information, especially as it is one place where we have radiocarbon and minerals determined in the same material.

Also, analysis of the mineral assemblage of the clay fractions is based entirely on X-ray diffraction with Cu Kα radiation. Thus, there is no proper estimate of possible contents of SRO phases and oxides of the clay fractions. Several sections of the manuscripts left me under the impression the authors are not aware that the clay fraction is not composed entirely of clay minerals but also contains other phases capable to interact with organic matter.

We do understand the point and have clarified throughout the text.

The density separation used is also rather strange. The cut-off density of 1.7 g/cm3 is not justified, at last not by the reference given. In addition, the sonication energy used was little, thus, total dispersion of samples with stable aggregates has to be questioned. Especially, the more oxide and kaolinite-rich samples may not disperse completely, thus, the heavy fraction likely still may contains light fraction-type material. In turn, undispersed aggregates may still contain enough light material to make them float.

We decided to measure only the so-called 'free' light fraction, i.e. the material that floats but not the material that requires strong sonication to destroy aggregates. The cutoff density of 1.7 g cm$^{-3}$ is one that means most SRO minerals (i.e. lowest density mineral phases) will

not float.   Density separation is a technique that is adapted to the soils used and there is not really a standard protocol.  It is very clear that the procedure used definitely influences the results (please see Castanha et. al, 2013 who discuss this in detail).  As we used a common procedure for all samples, we assume that results can be compared within our study, though care must be taken when comparing with other studies that may have used other methods. .

The rather variable and often very small contents of organic carbon in the light fractions point at major problems with the separation. In summary, the density separation has been carried out in a hardly reproducible manner.

Often in clay-rich soils, there are some mineral phases that are attached to low density material, or that remain floating in the sodium polytungstate solution even after a very long time of centrifugation.  Some of these can be siphoned on to the filter when removing the floating organic matter.   Including the total weight of these phases and the C content is important for determining the yield of the procedure, and reported low C contents are not uncommon, especially in B horizons and (in our data set) in clay-rich soils.   The presence of small amounts of mineral materials on the filter can dilute the C content overall but have a negligible effect on the C isotope signature.   For example if 50% of the weight of isolated material is mineral-dominated with a concentration of 0.5%C, and the other 50% of the weight is free organic C with 40%C.   then the overall %C of the mixture on the filter would be 20.25%, a large dilution.  However if we combust and analyze the isotopic signature of the mixture, the part of the mixture with 0.5%C would contribute 0.5/25.25, or about 2%, of the total C in the sample.    In the case of the basalt soils (which averaged about 10%C, the contribution from the mineral-associated C could be higher (in our example, .5/10, or 5%).   Using our own data as an example, assuming [14]C signatures of free organic matter of 1.100 fraction modern and 0.8 fraction modern for the mineral-C,  the total fraction modern we measure on the mixture would be 1.096 (instead of 1.100),  and reduces the presumed TT by 1-2 years  (either from 10 to 9 years, or 65 to 63 years).

For this reason, we are not concerned by the degree of dilution in reporting our isotopic signatures for the root-free fLF fraction.  In a revision, we will point out that %C results are subject to uncertainties in the root-free fLF fraction due to the potential inclusion of mineral material on the filters.

Minor concerns:
Title: As written, the title suggests a study of only local importance. The authors may consider changing it into "Timescales of C turnover in soils with mixed crystalline mineralogies".

p.1, l. 17-18: The conclusion that the enrichment of OC in the clay fraction is due to stabilisation by clays is either trivial (in case of the authors refer to clay as size fraction) or disputable (in case of referring to clay minerals; see above and below).

We choose the disputable and will clear up confusion throughout the text as detailed below.

p. 1, l. 23: What made the authors believe that crystalline Al oxyhydroxides contributed to the accumulation of faster turning over fraction? Is that based on DCB-extractable Al? If yes, please note that the extractant is not capable of extracting much Al from crystalline Al phases such as gibbsite.

We recognize the confusion created here and note for the record that we do not believe that the DCB extractable Al is a useful indicator of mineral composition. We will remove Al from that sentence, and have generally removed Al(d) from the Tables as well (it is still given in the supplementary material).

p. 1, l. 26: What does SRO refer to? Seemingly it refers to the observed relation- ship between DCB-extractable Fe and organic carbon. Dithionite-citrate-bicarbonate-extractable Fe includes Fe from crystalline Fe oxides, such as goethite and haematite.

As this reviewer points out the definition of SRO minerals is vague. Here we use the standard approach of evaluating the oxalate extract for the amounts of Fe and Al released during extraction. We assume that Fe release is due to decomposition of ferrihydrite or perhaps nano-crystalline goethite and that the Al release is due to decomposition of nano-crystalline aluminosilicates such as allophane and imogolite. We recognize that we may not have been clear about this operational definition and will clarify throughout the text.

p.1, l. 29-30 (and elsewhere in the manuscript): Expressions such as "crystalline Fe" are nonsense and misleading. Iron is an element. Please refer to the correct mineral phase, e.g., Fe oxides.

The reviewer is correct. We regret the sloppy short hand that crept into our text; it should be fixed now.

p. 2, l. 3-5: Some references given do not refer to physical protection or recalcitrance. Please be re-consider.

We completely rewrote and simplified the text in the introduction based on this and several other comments by the reviewer below.

p. 2, l. 7-16: Somehow, I am under the impression the authors have some problems with mechanisms involved in organic-mineral interactions. For example, what is "de- hydration bonding"? Also, the choice of references is a bit strange. For instance, two references are on methods for estimating mineral phases but do not address binding mechanisms. Did Masiello et al. (2004) really refer to crystalline Al and Fe sequioxides?

The reviewer is justified in not understanding our highly compressed text which tried to cover too much ground in a short space – something we think is not really important for the paper anyway, and therefore was rewritten to achieve greater simplicity and clarity.

p. 2, l. 10: Replace "metallic" by "metal".

Done.

p. 3, l. 13-16: Note, the studies of Wattel-Koekoek et al. (2003, 2004) include no proper estimates of possible contents of oxides, thus, do not allow for distinguishing of effects by clay minerals and other phases, including SRO phases.

The reviewer is strictly correct in this point although those authors selected samples to analyze that ensured a dominance of crystalline alumino-silicate clays with or without crystalline Fe oxides.

p. 3, l. 17-23: I suggest adding proper research questions and/or hypotheses.

We have re-written this section and now make the questions more explicit.

p. 3, l. 27-28: What is "soil residence time"? Please explain.

The Kruger sampling sites offer a unique landscape for soil sampling. All the streams that cross the park from west to east are maintained at the same erosional base level by a strata of rhyolite that is much more resistant to erosion than the granites and other volcanics that are upstream of it. Furthermore we are able to establish erosion rates on the granites using [10]Be accumulated in quartz sampled in this case from stream channels sands. As we state in the referenced citation (Chadwick et al., 2013): "Using average regolith depth and catchment-averaged erosion rate estimates, we infer long hillcrest regolith residence times of 0.11, 0.15, and 0.57 m.y. for the dry, intermediate, and wet sites, respectively." These data are corroborated by measured soil production rates (Heimsath et al. in prep.). The importance of the stream channel base level control is that it means that all landscapes regardless of whether they are underlain by granite or volcanic rocks are eroding at the same overall rate. This gives us confidence that the soil landscape is highly stable as one would expect for a craton in a non-glaciated environment and as a consequence mineral transformations can be expected to have moved past the meta-stable SRO stage toward a stable end product (given a specific climate condition). This approach to sample selection was also used by Wattel-Koekoek et al. (2003, 2004), except they used it a more global context without a specific local landscape context.

p. 4, l. 24-25: Please give more information on the determination of the clay contents. Did the procedure involve pre-treatments such as destruction of organic matter and oxides?

As mentioned in the text, we used $H_2O_2$ that destroyed part of the organic matter in the clay fraction that was measured by XRD.

p. 4, l. 32-33: The location of the CN analyser is probably of no importance. Omit.

Done.

p. 5, l. 3: As stated, it seems only the characterisation of clay minerals was attempted.

The reviewer is correct that "as stated" it appears that only characterization of clay minerals was attempted. There are several parts to that statement. First we specifically did not characterize sand and silt size mineralogies. We did use a separate approach to characterize the SRO minerals and Fe oxyhydroxide minerals. For these we conducted oxalate and DCB extractions on the <2-mm fine earth fraction. The reason for using the fine earth was that we were concerned that some of these minerals would be coating the sands and silts in ways that would be missed if we only conducted those extractions on the <2-µm (clay size) fraction. In the methods section we covered these extractions in the previous section on soil characterization which led to an artificial separation of the extraction quantification of the clay minerals from the XRD characterization.

We have attempted to rewrite the text to make it clear that we are relying on both the extractions and the XRD approaches to develop the quantitative understanding of the soil mineral composition. It should be noted that we recognize that mixing these approaches is not the best way to get a soil mineral compositions, but we also argue that there is no readily accepted single approach to full quantitative mineral characterization of soils. As a consequence, we are fully aware that our development of graphical relationships among mineral compositions and carbon turnover is flawed by our acceptance of specific operational approaches toward mineral quantification. However, we now keep the comparisons strictly between the same kinds of samples – e.g. bulk C and bulk TT versus the bulk Fe(d)-Fe(o) measures on the one hand, and C strongly associated with the Clay-sized fraction that was also used for XRD measuremnent of mineralogy.

p. 5, l. 14-16: What was the idea behind adding Al oxide to the clay fractions before X-ray diffractometry? Why not quartz? The addition of Al oxide limits estimation of Al oxide phases. Was the quantification of X-ray diffraction data supplemented by data on the elemental composition? What software was used?

The manuscript has been clarified to state that "corundum" was used as the XRD standard. Corundum has sharp peaks in XRD spectra that overlap with relatively few phases common in soil (including gibbsite) and these peaks degrade minimally during the grinding process used to mix sample and standard. Preliminary processing of the XRD spectra did not suggest gibbsite was an important constituent of the clay mineral fraction and gibbsite is not considered to be a major sorber of organic matter in soils. The word "software" has been added to clarify that the Rockjock software was used to for the quantification of minerals from XRD spectra.

p. 5, l. 24: Why a density-off of 1.7 g/cm3 was used? The reference given refers to 1.6 g/cm3.

The density of 1.6 g/cm$^3$ is typically below those of all SRO minerals; so is 1.7 g/cm$^3$.  We consulted with the author of the reference (Marion Schrumpf) about which density to use, and this was her suggestion.   There is no general agreement on methods to use for density separations and many different density cut-offs can be found in the literature.

p. 5, l. 25: What was the reason of using varying amounts of sample for the density separation? Why no standard protocol was used? Did the authors attempt estimating the carbon recovery?

The amount used was 10-15 grams, we did not feel the need to control the amount of sample extracted to better than within a few grams as the yield was determined based on the measured initial weight for each sample.   We did estimate C recovered in each fraction (these data are given in Supplementary Table 1, and indeed they are not as beautiful as we could wish (recovery based on adding the fractions together ranges from 40-95% for surface soils).  We are most confident of the % of total C in the HF-fraction as there are issues with weight change in filters and low masses with the quantification of the low-density fraction, and potentially loss of material when picking roots off of the filters).   An additional amount of C is dissolved and not recovered in the dense liquid.  We admit that our mass balance (as occurs in many density separation procedures) was not perfect.  However, as outlined above, we do not think this affected isotopic results – or at least it affected them in the same systematic ways.   Please see Castanha et al. (2008) for a detailed discussion of the various ways density fractions are affected by the procedures used.

*Castanha, C,  S Trumbore, R Amundson (2008)  Methods of separating soil carbon pools affect the chemistry and turnover time of isolated fractions
Radiocarbon, 50, 83-97.

p. 5, l. 26: What is meant with "ultrasonicated at 60 J mL-1 for 2.5. min"? Is 60 J/ml the total energy input? Did the authors control for proper dispersion, i.e., disaggregation? And if yes, how this was done?

60 J/mL is an estimate of the energy input, determined after calorimetrial calibration of the sonicator.  This is a relatively low energy and not likely to disrupt strong aggregates. Schrumpf et al. 2013 used stepwise increases in energy input to determine the level at which all aggregates were dispersed (we are using the identical system that she used). They found that   "Energy input of 100 J mL$^{-1}$ was sufficient"  (to destroy all aggregates) "for sandy soils (Bugac,  Bordeaux), and between 300 and 450 J mL$^{-1}$ for most other  soils. For the clay-rich Hainich soil, the energy input had to be raised to up to 900 J mL$^{-1}$) . Clearly we did not destroy all aggregates with this procedure, and this was not our intent. Thus our mineral fraction may include low density material that was protected in aggregates.   This is part of the general problem in such operationally defined fractionation methods, and one of the points of the paper is to explain the common observation that the heavy fraction is a mix of materials with different $^{14}C$ signatures.

p. 5, l. 29-30: What is the idea behind removing visible roots from the light fraction? Actually, dead roots are part of the debris material that makes up the light fraction. Why the authors did not remove living roots from fresh samples?

Castanha et al. (2008, reference above) demonstrated that the radiocarbon signature of the low density fraction is strongly affected by the presence of fine roots. Normally these are picked from samples as part of the sieving to <2mm; however, different people pick fine roots more or less diligently. Castanha et al. (2008) showed that picking the fine roots out of the low density fraction minimized variability among 'operators'. Also, we know (because we measured them) that the fine roots have mostly contemporary C, and wanted to know what the rest of the C in the low density fraction contained, especially as we would expect charred materials in these fire-prone regions.

p. 6, l. 1-4: Due to the bias in the clay and density separation I do not think the data set to be solid and comprehensive. I am wondering why the authors did not try a more logic approach, such as first separating all light material, then, separating the clay fraction from the heavy fraction.

Figure 4 was intended to be transparent about the overlaps between isolated fractions. While we agree with the reviewer that it might be more satisfying to have all fractions isolated sequentially so that there is no such overlap, this is not what we did. One reason for this is that the density separation is expensive, and did not always yield enough clay for the mineralogy step, especially in granites, where clay content was very low. We would definitely do this differently in the future, but cannot change the past.

p. 6, l 13-14: Is acidification to pH 6 really enough remove all carbonate, including that bonded to other mineral phases? I doubt.

Actually carbonates are remarkably non-bonded to other mineral phases and tend to reside in soil as their own unique bodies (K fabric concepts). We do expect that acidification to pH 6 will remove the carbonates although there is the possibility that some carbonates could avoid decomposition if protected within aggregates. As pointed out in the text the carbonate in the horizons sampled was primarily in relatively large aggregates (sand and pebble size) whereas the bulk of the fine-earth fraction was non-calcareous (did not react to acid in the field). We do point out the one place where carbonates in the ClayXRD fraction could play a role – in all other samples, there was no measureable carbonate in the samples analyzed for 14C and reported in Table 4.

p. 7, l. 13: No, DCB does not extract only crystalline Fe oxide phases but all Fe oxide phases, including ferrihydrite. In turn, oxalate extracts only the poorly crystalline portion of DCB-extractable Fe.

Yes the reviewer is correct: to get at the crystalline Fe oxide phase we subtract the oxalate extracted Fe from the DCB extracted Fe. We have pointed out in the text that Fe(oxides)=Fe(d)-Fe(o) and create new columns in the appropriate Tables with output of

that calculation; we have also made sure the Figures use the correct values as well (results do not change).

p. 7, l. 14: Note, DCB-extractable Al does not represent Al oxide phases.

Right the DCB-extracted Al is meaningless in this context – have removed that sentence.

p. 7, l. 18: Is there are reason why the clay contents increased with depth?

Increasing clay with depth through the solum is quite typical for soils due to hydrological transfer of colloids. Typically the downward transfer of colloids is countered by bioturbation which mixes profiles, but our observation is that more often than not soils have a subsurface accumulation of clay-size materials, often skewed to the small particle sizes.

p. 8, l. 27-32: Is it correct, the soils containing pedogenic carbonates are those rich in smectite? So, could it be that their radiocarbon signature was affected by carbonate? By the way, the picrite (back basalt)-derived soil is classified as Calciustert but not listed as containing carbonate? Isn't that illogic?

There was no evidence from the XRD data to suggest that the clay fraction harbored calcite. If the picrite soil was carbonate free or mostly so then the classification should be Typic Haplustert.  This has been corrected in Table 1.

p. 9, l. 8-9. Due to the H2O2 treatment of the clay fraction, I doubt that proper estimates on non-clay carbon are possible.

The problem here is with the definition of non-clay C.  We meant this to mean all C (including that oxidized by $H_2O_2$) that was not in the clay fraction measured for XRD.  We are more careful with this definition in the revised text.  The mass balance stands – the C removed included all non-clay sized material and all material removed by $H_2O_2$ from clay sized material.

p. 10, l. 9-15: These correlations need to be re-considered bearing in mind that the clay fraction contains most of the oxides.

We now make clear in the text that subtracting the Feo from the Fed prior to determining the crystalline Fe oxide concentrations.

p. 11, l. 9-17: The discussion here is rather speculative since composition of organic matter was not addressed.

We agree, but also feel that we did point out in the text that we were speculating on this – effectively connecting the dots from pieces of the literature and our measurements.

p. 11, l. 18-29: Here, it becomes clear the authors partly equal clay-sized particles and clay minerals. They relate the estimated non-clay fraction carbon to "other, non clay mineral stabilization mechanisms". This is simply wrong, since the clay fraction holds also most if not all oxide phases (even the Cu Kα X-ray diffraction, despite of being rather insensitive to Fe oxides, indicated their presence). Also, I have problems with the authors' logic. The organic carbon of all study soils relates well to the DCB-extractable Fe. Seemingly, the clay mineral type does not matter much to the accumulation of organic carbon. So, there is no reason for all the clay mineral discussion. I am also wondering, why the authors did not comment on the rather small contents of organic carbon in the smectite-rich soils. There is much writing on organic matter stabilisation by smectites. The results presented, however, suggest that smectites are rather poor organic carbon accumulators.

The only oxides quantified by XRD were Fe-bearing and included hematite, goethite, magnetite, maghemite and ilmenite. We agree that we create confusion when we also report data on crystalline Fe oxihydroxides based on the Fe(d) – Fe(o) for the bulk soil. In the plot below, we compare the DCB-oxalate Fe phases for the bulk soil (x- axis) to an upscaling of the Fe oxihydroxides measured with XRD corrected for the % of clay-sized material (y-axis). This plot is now given in the Supplemental Material as Figure 1. There is general correspondence, although we agree that the bulk extracts are a better measure since they also include things like coatings on sand or silt-sized materials. Also, we have far more data for the bulk extracts.

[Figure]

The clay minerals are important for the radiocarbon, where a small amount of old material has influence. The bulk of the C is stabilized by mechanisms that have timescales that yield similar 14C signatures. Something similar was found by Lawrence et al. (2015), so we thought it important to point this out.
.

By the way, why do the authors consider "crystalline Fe and Al (oxyhydr)oxides" as controlling carbon accumulation? The close relationship is for DCB-extractable Fe and not for any Al. Also remember, DCB extracts all non-crystalline Fe oxides (see above).

We understand the problem and increased clarity throughout the text.

Assuming that the relationship between organic carbon and Fe oxides is also valid for the smectite-rich soils, i.e., Fe oxides do the accumulation; it is absurd to assume that the smectites make the organic carbon turning over slowly. As already pointed out, I assume an issue with incomplete removal of carbonates to be the reason of the radiocarbon signature of the smectite-rich soils.

We disagree that carbonates can be responsible for the old ages in smectite clays, expecially as no carbonates were found in the clay-sized fraction using XRD. To obtain an age of 2000 radiocarbon years, roughly 20% of the carbon in the sample would have to be radiocarbon-free. As the calcites we measured were not radiocarbon free, they would have to make up an even larger portion of the total C measured for isotopes. Of the the smectite-rich clays in Table 2, all but one had <1% Carbonate. It is highly unlikely that inclusion of carbonates can be possible for the low $^{14}$C values we measured.

Tables and Figures (general): Please give proper dimensions. Percentages of smectite, organic carbon, smectite likely refer to weight-%. Consider giving instead g/kg. What is the dimension of the cation exchange capacity (CEC)?
None of the tables and figures is truly self-explanatory.

We have revised and hopefully this is now better.

Table 2: I am wondering if the "oxides" included also Al phases. The little clay content of the non-mafic soils is no good reason for not analysing the composition of their clay fractions.

The oxides presented here are for information only. We used the Fed-Feo data from the extractions to develop the Fe-oxide – carbon relationships because of concerns about loss of oxides (as coatings) during particle size separation.

Table 3: The rather low organic carbon contents of some of the light fractions support my concerns on issues with the density separation. The light fractions with little carbon may contain a good portion of mineral-bound (older) carbon, thus, suggesting a slow turnover.

Please see the answer to this issue above. While we agree that there are likely mineral-bound (older) C diluting the low density C, (a) we can not absolutely rule out that the C conents of this

fraction can be as low as 10%C and (b) dilution with mineral phases would contribute only a few per cent of the total C measured for isotopes, that would not have really big effects. We have not made strong interpretations of the fLF radiocarbon data in this paper, except to point out that they are different from the HF, and present them mostly for completeness.

Figure 5: Giving an enrichment factor for clay fractions treated with H2O2 is not valid.

We do not completely understand this comment. There is no 'enrichment factor' in the sense you would use for 13C isotopes. Radiocarbon data are all corrected for the 13C in the sample, and any such enrichment factors are corrected for. What is left is the mean age information that is given in the figure.

**Anonymous Referee #2**

GENERAL COMMENTS
Summary: This manuscript investigates the controls on carbon turnover and C inven- tory across soils ranging in mineralogy in Kruger National Park, South Africa. The authors sampled soil across a variety of parent materials resulting in differing min- eralogical characteristics. Older soils with low contents of short-range order mineral phases were chosen to specifically test the impact of phyllosilicate clays on C turnover time and stocks. To elucidate mineral protection mechanisms, particle size and density fractionation was used in combination with 13C and 14C measurements. The main finding is that the content of high surface area, 2:1 phyllosilicates (i.e., smectite) is a better predictor for C turnover times in these soils than clay content (i.e., < 2 µm size fraction). They conclude that analysis of phyllosilicate clay composition reveals clearer insights into C stabilization mechanisms than clay content alone.

Strengths: This manuscript provides a unique dataset in that it is one of very few that measured radiocarbon on different soil fractions (particle size and density) of the same soils. This approach puts the authors in a position to examine the usefulness of individual fractionation approaches for different soil system, and provides interesting insights into the mineral protection mechanisms responsible for C storage in low-SRO, phyllosilicate clay dominated soil systems. This manuscript will be of broader interest to the SOIL readership and I support its publication.

Weaknesses: 1) The authors don't draw a clear line between two definitions for 'clay', i.e. clay as a particle size category and clay as phyllosilicates. It would be helpful to be consistent with this terminology throughout the manuscript
2) The manuscript could use some editing. The introduction is not very concise and could state the research question more clearly. There are typos, somewhat confusing sentence structures and word choices, and mislabeled figures throughout (see specific comments below).

We have rewritten the introduction to make it more concise and to the point and will clarify the use of the term "clay" throughout.

SPECIFIC COMMENTS
Abstract
P1L18: What does 'stabilized' mean in this context? Adsorbed?
We use stabilized to refer to C that is retained in the soil without reference to any specific mechanism or timescale.  However, we agree with the reviewer and have tried to remove this word throughout the text.

P1L19-20: This is a speculative argument informed by the data presented here and should be framed as such. It is a valid point to make, but it should not be framed as a result.
Agreed, we have changed this.

P1L21-24: This argument is confusing to me. The authors state that the fraction that is not clay (> 2 µm) has much shorter turnover time than the clay fraction. They then attribute the shorter TT in the > 2 µm fraction to weaker bonding of C to Fe and Al oxyhydroxides and kaolinite as well as the presence of more modern plant inputs (lightfraction). The part that's confusing to me here and throughout the manuscript is the fact that I would consider Fe/Al oxyhydroxides and kaolinite as clays. They could be clays either because of their small size (oxyhydroxides) or because they are phyllosilicates (kaolinite). I understand the authors' point in the discussion that some of these minerals may have stuck to larger grains and so were removed with the > 2 µm fraction. But I strongly suggest making sure you clearly separate when you talk about clays as size separates and when 'clay' means phyllosilicate.

All the above points have been incorporated into a revision of the introductory text material, the discussions above hopefully help clarify this.

L24: HF and LF are not defined
Thank you, this is fixed.

P2L8: What is "dehydration bonding"? Do you mean ligand exchange?

Removed from the introduction.

P2L13-15: Poch et al. seems an odd reference here. It's work on clays on Mars.

We will remove this reference.

P2L15-16: Masiello et al found correlations between pyrophosphate extractable Fe and Al and turnover time. I thought that pyrophosphate extracts poorly or amorphous Fe and Al phases, not crystalline sesquioxides as the authors state here.

Yes it is correct that it does not extract crystalline sesquioxides – probably more likely Fe and Al oxides bound to organic ligands. Actually the line between pyrophosphate and oxalate extracted material is pretty fuzzy.

P2L17-24: This argument seems very convoluted to me. Isn't the argument that organics on different mineral phases may exchange at different rates and therefore have different turnover times. Fractionation techniques average across a number of these interactions, and so there is a need to look into mineral composition of these fractions more closely to gain insights into what mineral phases provides the most protection (and thus the longest turnover time)? I see where the authors are headed with this argument, but this paragraph could be more concise.

We have rewritten the introduction

P2L28: What is 'older C storage'. Choice of words (c.o.w.) is odd.

Agreed, we have changed this.

P2L32: Lawrence et al provide a large dataset, but it's not true they 'virtually' measured 'all' reactive components in the system.

We concede that point although we do think that they did better than most papers when it comes to field-based sampling, lab characterization and correlation analysis of the results.

P3L2-4: These two sentences makes little sense to me. I think it would help to frame the argument in terms of C stocks and turnover times, rather than 'storage' and 'stability'. That applies to the remainder of the paragraph.
Thank you for this suggestion, we have tried to adopt this throughout the manuscript.

P3L8: 'Trade-offs" of what?

The idea relates back to the sentence starting on the first line of the page, but we recognize that there were a long few lines between the two points – we will clarify in a rewrite.

P3L9-10: I agree with this statement, but that is not a research question.

It was not meant to be a research question but we can understand why the reviewer would be yearning for a pithy purpose statement by now.  We now have identified 3 research questions.

P3L10: Having this very broad goal stated here seems a bit misplaced. I think the authors should state the specific research question here.

Right.

Methods

P3L28: 105 yrs? I might be wrong, but shouldn't it be longer?

It was a typo, should be 10,000 (i.e. $10^5$)

P4L31: Typo, should be Plasma

Thanks

Results

P7L19-20: where does it go up to 26%. The values in Table 2 range from 0-15%?

In Table 2, the % smectite (S) column has 23% and 26% for GR-550-T soils. The column labeled O (oxides) is the one that ranges from 0-15%. What is written in the text is, as far as we can tell, consistent with the Table.

Table 2 was also published previously, better in the SI?

Only three of the samples (GR-550) in Table 2 were published in Khomo et al. – we would prefer to keep the new data in the text; if the reviewers feels some of our tables can be moved to supplementary material, we will gladly do this.

Table 3: I would include total C in HF (mass of C). There are also a few typos in the table header.

We have fixed typos but prefer not to put mass of C in the text because we know there are such large issues with gravel content in many of our sites. These data are given in Supplemental Table 1 and used for profile-average calculations.

P9L7: I think the authors are referring to Fig. 5. The authors should check what figures are referred to in the following. It didn't always seem to match up.

Done.

P9L21-29: Why does every sentence refer to Fig. 4? I'm not sure what is said here refers to anything I can see in Fig. 4.

We felt that Figure 4 was crucial to provide clarity about the overlapping nature of the organic C fractions we measured. The idea was to illustrate how these varied with one sample, but we have tried to remove extraneous references.

P9L23ff. On the issue of whether or not H2O2 oxidizes mineral bound OM. I think it does, and it's a mute point to argue it doesn't. But the authors could at least find some refs to support the claim that it has minor impacts.

We agree that $H_2O_2$ oxidizes some mineral bound OM, and did not mean to imply that it did not. Our point was that a lot of C still remained, as most of the clay samples treated with $H_2O_2$ still had organic C concentrations of 1-2%.

Discussion
P10L27: c.o.w. What are 'C properties'?
We have changed this.

P12L6: c.o.w. 'sorbers' is not a word. It's 'sorbents'.
Agreed, and changed.

[revised manuscript text omitted]

Comment Subject: Font color: R,G,B (0,0,10), Don't allow hanging punctuation

| Page 1: [2] Style Definition | Susan Trumbore | 10/20/16 11:13:00 |

Comment Text: Don't allow hanging punctuation

| Page 1: [3] Style Definition | Susan Trumbore | 10/20/16 11:13:00 |

Footer: Font color: R,G,B (0,0,10), Suppress line numbers, Don't allow hanging punctuation

| Page 1: [4] Style Definition | Susan Trumbore | 10/20/16 11:13:00 |

Equation: Font color: R,G,B (0,0,10), Don't allow hanging punctuation

| Page 1: [5] Style Definition | Susan Trumbore | 10/20/16 11:13:00 |

Balloon Text: Font color: R,G,B (0,0,10), Don't allow hanging punctuation

| Page 1: [6] Style Definition | Susan Trumbore | 10/20/16 11:13:00 |

List Paragraph: Font color: R,G,B (0,0,10), Don't allow hanging punctuation

| Page 1: [7] Style Definition | Susan Trumbore | 10/20/16 11:13:00 |

Copernicus_Word_template: Font color: R,G,B (0,0,10), Don't allow hanging punctuation

| Page 1: [8] Style Definition | Susan Trumbore | 10/20/16 11:13:00 |

Header: Font color: R,G,B (0,0,10), Suppress line numbers, Don't allow hanging punctuation

| Page 1: [9] Style Definition | Susan Trumbore | 10/20/16 11:13:00 |

Bullets: Font color: R,G,B (0,0,10),  No bullets or numbering, Don't allow hanging punctuation

| Page 1: [10] Style Definition | Susan Trumbore | 10/20/16 11:13:00 |

Betreff: Font color: R,G,B (0,0,10), Don't allow hanging punctuation

| Page 1: [11] Deleted | Susan Trumbore | 10/20/16 11:13:00 |

carbon in a field setting designed to minimize the role of SRO by taking advantage of multiple lithologies

| Page 1: [12] Deleted | Susan Trumbore | 10/20/16 11:13:00 |

material had average TT of 1020 ± 460 years in surface soils. The mean TT of this clay-associated C increased with depth and with fraction of

clay

**Page 1: [13] Deleted**                        Susan Trumbore                        10/20/16 11:13:00 PM

was smectite. Because the C associated with smectite clay was so old, the amount of smectite (2:1 clays) controlled the age of

**Page 1: [14] Deleted**                        Susan Trumbore                        10/20/16 11:13:00 PM

Kruger landscapes. The TT of the majority of soil C –not stabilized by clays - was much shorter, $190\pm190$ years in surface horizons. We suggest that this faster component reflects timescales of weaker C stabilization by crystalline Fe and Al oxyhydr)oxides and kaolinite (1:1) clays, as well as LF fractions not associated with minerals. Thus, bulk or HF carbon integrates C stabilized by mechanisms with inherently different TT, something that is often inferred from radiocarbon measurements. While SRO mineral concentrations were very low in these soils, the soils with most SRO had very high C content but also very young C. In other environments, SRO can be very stable and sorb C on very long timescales. We hypothesize that the seasonal wetting and drying in the KNP may reduce the age of SRO minerals as well as the C

**Page 1: [15] Deleted**                        Susan Trumbore                        10/20/16 11:13:00 PM

them. Across the varying lithologies and a precipitation gradient found in the KNP, we found mineralogy to be the most important explanatory factor for C content (related to crystalline Fe) and turnover time (related to the amount of smectite).

**Page 2: [16] Deleted**                        Susan Trumbore                        10/20/16 11:13:00 PM

range of interactions with inorganic compounds, but in the complex soil medium our understanding of the contributions of each mechanism is limited (von Lützow et al., 2007). For example, organic ligands can bond with trivalent metal ions such as iron (Fe) and aluminum (Al) or by dehydration bonding with metallic nano-(oxyhydr)oxides (Parfitt and Childs, 1988; Kaiser and Zech, 1996; Chorover et al., 2004). At high clay concentrations and soil surface area, organic ligands can be sorbed into the matrix of short-range-order (SRO) nanocrystalline aluminosilicates or metallic (oxyhydr)oxides (Torn et al., 1997; Kaiser and Guggenberger, 2003; Chorover et al., 2004; Kramer et al., 2012) or onto crystalline silicate clay minerals (Kaiser and Guggenberger, 2000; Sollins et al., 2009). However

**Page 2: [17] Deleted**                        Susan Trumbore                        10/20/16 11:13:00 PM

In addition, crystalline pedogenic Fe and Al sesquioxides are associated with C that has ages ranging from hundreds to thousands of years (Masiello et al., 2004).

Soil C can be operationally separated into a light fraction not

**Page 2: [18] Deleted**                        Susan Trumbore                        10/20/16 11:13:00 PM

(low density fraction; LF) and a heavy fraction associated with minerals (high density fraction; HF). Decomposition of the light fraction may be governed by interactions among decomposers and C substrate quality, as well as by occlusion within mineral aggregates. Stabilization mechanisms in HF can operate on a range of timescales (e.g.

**Page 2: [19] Moved to page 2 (Move #1)**          **Susan Trumbore**          **10/20/16 11:13:00 PM**

Schrumpf and Kaiser 2014; Schrumpf et al. 2013; Wattel-Koekkoek and Buurman 2004).

**Page 2: [20] Deleted**          **Susan Trumbore**          **10/20/16 11:13:00 PM**

Given the different strengths of mineral-C interactions, there is a need to measure carbon storage and age of C within the dense fraction of mixed mineralogy soils where the quantity of C-sorbing compounds including clay minerals has also been measured. With this information, we can attempt to predict C storage and timescales of stabilization from known amounts of mineral constituents (e.g. 2:1 versus 1:1 clays) or from readily available proxy measurements such as surface area or cation exchange capacity (Lawrence et al., 2015

**Page 2: [21] Deleted**          **Susan Trumbore**          **10/20/16 11:13:00 PM**

One approach to evaluate mineral controls on C storage and turnover is to select samples for analysis from distinctly different global environments to cover a suite of mineral compositions. For example, Wattel-Koekkoek et al. (2003) used this approach to document that larger quantities of crystalline 2:1 clays were associated with greater (and older) C storage, while 1:1 clays were not. Another approach is to sample soils along gradients of state factors to isolate specific soil property gradients and evaluate correlations among varying soil properties and C properties. The latter approach has proved useful (e.g. Trumbore et al., 1996; Torn et al., 1997; Masiello et al., 2004) but often without full quantification of all the possible controls on C storage. By contrast Lawrence et al. (2015) quantified virtually all reactive components along a humid, forested chronosequence formed on alluvium from andesitic volcanics and evaluated their correlations with C storage and turnover. They found a complex suite of controls for C storage and age related to depth-dependant C inputs and depth-dependant inorganic chemical properties. In surface horizons, they found increases in C storage but not its stability with increasing amounts of pyrophosphate extractable Fe and Al. In subsurface horizons they found increased C stability, but not increased storage, with increases in surface area and halloysite clay concentrations. In contrast with other studies cited above, they found little correlation between C storage or stability and the amount of smectite clays or SRO minerals. Although it would be ideal to be able to make predictions about C storage and stability based on chemical and mineralogical properties, this (Lawrence et al. 2015) and other studies (e.g. Masiello et al., 2004; Wattel-Koekoek et al., 2003; Wattel-Koekoek et al., 2004) indicate that there may be significant C-storage trade-offs occurring within soils of differing mineral compositions. Therefore we need more case studies that cover a broad range of environmental, ecological and soil chemical conditions. Our long-term goal is to refine understanding of the sensitivity of mineral-associated organic matter to climate and land-use change.

Few studies have focused

**Page 5: [22] Deleted**          **Susan Trumbore**          **10/20/16 11:13:00 PM**

the bulk soil. Experience has shown the greatest contribution from LF to bulk soil C in A horizons; in deeper horizons, we did not perform density separations and instead assumed that the bulk fraction approximates HF for properties such as C and C isotope concentrations.

| Page 8: [23] Deleted | Susan Trumbore | 10/20/16 11:13:00 PM |
|---|---|---|

%)

| Page 8: [23] Deleted | Susan Trumbore | 10/20/16 11:13:00 PM |
|---|---|---|

%)

| Page 8: [23] Deleted | Susan Trumbore | 10/20/16 11:13:00 PM |
|---|---|---|

%)

| Page 8: [23] Deleted | Susan Trumbore | 10/20/16 11:13:00 PM |
|---|---|---|

%)

| Page 8: [23] Deleted | Susan Trumbore | 10/20/16 11:13:00 PM |
|---|---|---|

%)

| Page 8: [23] Deleted | Susan Trumbore | 10/20/16 11:13:00 PM |
|---|---|---|

%)

| Page 8: [23] Deleted | Susan Trumbore | 10/20/16 11:13:00 PM |
|---|---|---|

%)

| Page 8: [23] Deleted | Susan Trumbore | 10/20/16 11:13:00 PM |
|---|---|---|

%)

| Page 8: [23] Deleted | Susan Trumbore | 10/20/16 11:13:00 PM |
|---|---|---|

%)

| Page 8: [23] Deleted | Susan Trumbore | 10/20/16 11:13:00 PM |
|---|---|---|

%)

| Page 8: [24] Deleted | Susan Trumbore | 10/20/16 11:13:00 PM |
|---|---|---|

kaolins

| Page 8: [24] Deleted | Susan Trumbore | 10/20/16 11:13:00 PM |
|---|---|---|

kaolins

| Page 8: [24] Deleted | Susan Trumbore | 10/20/16 11:13:00 PM |
|---|---|---|

kaolins

| Page 8: [24] Deleted | Susan Trumbore | 10/20/16 11:13:00 PM |

kaolins

| Page 8: [24] Deleted | Susan Trumbore | 10/20/16 11:13:00 PM |

kaolins

| Page 8: [24] Deleted | Susan Trumbore | 10/20/16 11:13:00 PM |

kaolins

| Page 8: [24] Deleted | Susan Trumbore | 10/20/16 11:13:00 PM |

kaolins

| Page 8: [24] Deleted | Susan Trumbore | 10/20/16 11:13:00 PM |

kaolins

| Page 8: [25] Deleted | Susan Trumbore | 10/20/16 11:13:00 PM |

(

| Page 8: [25] Deleted | Susan Trumbore | 10/20/16 11:13:00 PM |

(

| Page 8: [26] Deleted | Susan Trumbore | 10/20/16 11:13:00 PM |

LF

| Page 8: [26] Deleted | Susan Trumbore | 10/20/16 11:13:00 PM |

LF

| Page 8: [26] Deleted | Susan Trumbore | 10/20/16 11:13:00 PM |

LF

| Page 8: [26] Deleted | Susan Trumbore | 10/20/16 11:13:00 PM |

LF

| Page 8: [26] Deleted | Susan Trumbore | 10/20/16 11:13:00 PM |

LF

| Page 8: [26] Deleted | Susan Trumbore | 10/20/16 11:13:00 PM |
|---|---|---|

LF

| Page 8: [26] Deleted | Susan Trumbore | 10/20/16 11:13:00 PM |
|---|---|---|

LF

| Page 8: [26] Deleted | Susan Trumbore | 10/20/16 11:13:00 PM |
|---|---|---|

LF

| Page 8: [26] Deleted | Susan Trumbore | 10/20/16 11:13:00 PM |
|---|---|---|

LF

| Page 8: [27] Deleted | Susan Trumbore | 10/20/16 11:13:00 PM |
|---|---|---|

LF

| Page 8: [27] Deleted | Susan Trumbore | 10/20/16 11:13:00 PM |
|---|---|---|

LF

| Page 8: [27] Deleted | Susan Trumbore | 10/20/16 11:13:00 PM |
|---|---|---|

LF

| Page 8: [27] Deleted | Susan Trumbore | 10/20/16 11:13:00 PM |
|---|---|---|

LF

| Page 9: [28] Deleted | Susan Trumbore | 10/20/16 11:13:00 PM |
|---|---|---|

Depth profiles of

| Page 10: [29] Deleted | Susan Trumbore | 10/20/16 11:13:00 PM |
|---|---|---|

**Clay-**

| Page 10: [29] Deleted | Susan Trumbore | 10/20/16 11:13:00 PM |
|---|---|---|

**Clay-**

| Page 10: [30] Deleted | Susan Trumbore | 10/20/16 11:13:00 PM |
|---|---|---|

C

| Page 10: [30] Deleted | Susan Trumbore | 10/20/16 11:13:00 PM |

C

| Page 10: [30] Deleted | Susan Trumbore | 10/20/16 11:13:00 PM |

C

| Page 10: [30] Deleted | Susan Trumbore | 10/20/16 11:13:00 PM |

C

| Page 10: [30] Deleted | Susan Trumbore | 10/20/16 11:13:00 PM |

C

| Page 10: [30] Deleted | Susan Trumbore | 10/20/16 11:13:00 PM |

C

| Page 10: [30] Deleted | Susan Trumbore | 10/20/16 11:13:00 PM |

C

| Page 10: [30] Deleted | Susan Trumbore | 10/20/16 11:13:00 PM |

C

| Page 10: [30] Deleted | Susan Trumbore | 10/20/16 11:13:00 PM |

C

| Page 10: [31] Deleted | Susan Trumbore | 10/20/16 11:13:00 PM |

-associated and non-clay associated, we

| Page 10: [31] Deleted | Susan Trumbore | 10/20/16 11:13:00 PM |

-associated and non-clay associated, we

| Page 10: [31] Deleted | Susan Trumbore | 10/20/16 11:13:00 PM |

-associated and non-clay associated, we

| Page 10: [32] Deleted | Susan Trumbore | 10/20/16 11:13:00 PM |

- $F_{clay} \times \%C_{clay}$

| Page 10: [32] Deleted | Susan Trumbore | 10/20/16 11:13:00 PM |

- $F_{clay} \times \%C_{clay}$

| Page 10: [33] Deleted | Susan Trumbore | 10/20/16 11:13:00 PM |
|---|---|---|

$F_{clay} = (\%C_{clay} \times \%clay$

| Page 10: [33] Deleted | Susan Trumbore | 10/20/16 11:13:00 PM |
|---|---|---|

$F_{clay} = (\%C_{clay} \times \%clay$

| Page 10: [34] Deleted | Susan Trumbore | 10/20/16 11:13:00 PM |
|---|---|---|

with mostly smectite, $F_{clay}$ stabilized

| Page 10: [34] Deleted | Susan Trumbore | 10/20/16 11:13:00 PM |
|---|---|---|

with mostly smectite, $F_{clay}$ stabilized

| Page 10: [34] Deleted | Susan Trumbore | 10/20/16 11:13:00 PM |
|---|---|---|

with mostly smectite, $F_{clay}$ stabilized

| Page 10: [34] Deleted | Susan Trumbore | 10/20/16 11:13:00 PM |
|---|---|---|

with mostly smectite, $F_{clay}$ stabilized

| Page 10: [34] Deleted | Susan Trumbore | 10/20/16 11:13:00 PM |
|---|---|---|

with mostly smectite, $F_{clay}$ stabilized

| Page 10: [34] Deleted | Susan Trumbore | 10/20/16 11:13:00 PM |
|---|---|---|

with mostly smectite, $F_{clay}$ stabilized

| Page 10: [34] Deleted | Susan Trumbore | 10/20/16 11:13:00 PM |
|---|---|---|

with mostly smectite, $F_{clay}$ stabilized

| Page 10: [35] Deleted | Susan Trumbore | 10/20/16 11:13:00 PM |
|---|---|---|

'bomb' $^{14}C$ $\Delta^{14}C > 0‰$); i.e. dominated by

| Page 10: [35] Deleted | Susan Trumbore | 10/20/16 11:13:00 PM |
|---|---|---|

'bomb' $^{14}C$ $\Delta^{14}C > 0‰$); i.e. dominated by

| Page 10: [35] Deleted | Susan Trumbore | 10/20/16 11:13:00 PM |
|---|---|---|

'bomb' $^{14}C$ $\Delta^{14}C > 0‰$); i.e. dominated by

| Page 10: [36] Deleted | Susan Trumbore | 10/20/16 11:13:00 PM |
|---|---|---|

The

| Page 10: [36] Deleted | Susan Trumbore | 10/20/16 11:13:00 PM |
|---|---|---|

The

| Page 10: [36] Deleted | Susan Trumbore | 10/20/16 11:13:00 PM |
|---|---|---|

The

| Page 10: [36] Deleted | Susan Trumbore | 10/20/16 11:13:00 PM |
|---|---|---|

The

| Page 10: [36] Deleted | Susan Trumbore | 10/20/16 11:13:00 PM |
|---|---|---|

The

| Page 10: [36] Deleted | Susan Trumbore | 10/20/16 11:13:00 PM |
|---|---|---|

The

| Page 10: [37] Deleted | Susan Trumbore | 10/20/16 11:13:00 PM |
|---|---|---|

clay-associated C (oldest)

| Page 10: [37] Deleted | Susan Trumbore | 10/20/16 11:13:00 PM |
|---|---|---|

clay-associated C (oldest)

| Page 10: [38] Deleted | Susan Trumbore | 10/20/16 11:13:00 PM |
|---|---|---|

-C

| Page 10: [38] Deleted | Susan Trumbore | 10/20/16 11:13:00 PM |
|---|---|---|

-C

| Page 12: [39] Deleted | Susan Trumbore | 10/20/16 11:13:00 PM |
|---|---|---|

in soil forming factors and thus a range of mineralogy, our goal was

| **Page 12: [40] Deleted** | **Susan Trumbore** | **10/20/16 11:13:00 PM** |

Our initial hypothesis was

| **Page 12: [41] Deleted** | **Susan Trumbore** | **10/20/16 11:13:00 PM** |

clay

clay

| Page 12: [41] Deleted | Susan Trumbore | 10/20/16 11:13:00 PM |
|---|---|---|

clay

| Page 12: [41] Deleted | Susan Trumbore | 10/20/16 11:13:00 PM |
|---|---|---|

clay

| Page 12: [41] Deleted | Susan Trumbore | 10/20/16 11:13:00 PM |
|---|---|---|

clay

| Page 12: [41] Deleted | Susan Trumbore | 10/20/16 11:13:00 PM |
|---|---|---|

clay

| Page 12: [41] Deleted | Susan Trumbore | 10/20/16 11:13:00 PM |
|---|---|---|

clay

| Page 12: [41] Deleted | Susan Trumbore | 10/20/16 11:13:00 PM |
|---|---|---|

clay

| Page 12: [41] Deleted | Susan Trumbore | 10/20/16 11:13:00 PM |
|---|---|---|

clay

| Page 12: [41] Deleted | Susan Trumbore | 10/20/16 11:13:00 PM |
|---|---|---|

clay

| Page 12: [41] Deleted | Susan Trumbore | 10/20/16 11:13:00 PM |
|---|---|---|

clay

| Page 12: [41] Deleted | Susan Trumbore | 10/20/16 11:13:00 PM |
|---|---|---|

clay

| Page 12: [42] Deleted | Susan Trumbore | 10/20/16 11:13:00 PM |
|---|---|---|

ancient

| Page 12: [42] Deleted | Susan Trumbore | 10/20/16 11:13:00 PM |
|---|---|---|

ancient

| Page 12: [43] Deleted | Susan Trumbore | 10/20/16 11:13:00 PM |
|---|---|---|

Silicate clays

Silicate clays

| **Page 12: [43] Deleted** | **Susan Trumbore** | **10/20/16 11:13:00 PM** |
Silicate clays

| **Page 12: [43] Deleted** | **Susan Trumbore** | **10/20/16 11:13:00 PM** |
Silicate clays

| **Page 12: [43] Deleted** | **Susan Trumbore** | **10/20/16 11:13:00 PM** |
Silicate clays

| **Page 12: [43] Deleted** | **Susan Trumbore** | **10/20/16 11:13:00 PM** |
Silicate clays

| **Page 12: [43] Deleted** | **Susan Trumbore** | **10/20/16 11:13:00 PM** |
Silicate clays

| **Page 12: [43] Deleted** | **Susan Trumbore** | **10/20/16 11:13:00 PM** |
Silicate clays

| **Page 12: [43] Deleted** | **Susan Trumbore** | **10/20/16 11:13:00 PM** |
Silicate clays

| **Page 12: [43] Deleted** | **Susan Trumbore** | **10/20/16 11:13:00 PM** |
Silicate clays

| **Page 12: [43] Deleted** | **Susan Trumbore** | **10/20/16 11:13:00 PM** |
Silicate clays

| **Page 12: [43] Deleted** | **Susan Trumbore** | **10/20/16 11:13:00 PM** |
Silicate clays

| **Page 14: [44] Deleted** | **Susan Trumbore** | **10/20/16 11:13:00 PM** |

While depth-related increases in C TT may be partly controlled by changes in the age of C associated with the millenial C fraction and with changes in the relative amount of C stabilized on smectite versus sesquioxides, much more

| Page 16: [45] Deleted | Susan Trumbore | 10/20/16 11:13:00 PM |
|---|---|---|

Kaiser, K. and Guggenberger, G.: Mineral surfaces and soil organic matter. European Journal of Soil Science 54, 219-236, 2003.

Kaiser, K. and Zech, W.:Defects in the estimation of aluminum in humus complexes of podzolic soils by pyrophosphate extraction, Soil Sci. 161, 452-458, 1996.

| Page 24: [46] Deleted | Susan Trumbore | 10/20/16 11:13:00 PM |
|---|---|---|

| Page 26: [47] Deleted | Susan Trumbore | 10/20/16 11:13:00 PM |
|---|---|---|

| Page 26: [48] Deleted | Susan Trumbore | 10/20/16 11:13:00 PM |
|---|---|---|

Section Break (Next Page)

[Figure]

[Figure]

[Figure]

[Figure]

Figure 4.

[Figure]

[Figure]

---

## Author Response (AR2)

Dear Editor,

Thank you for the chance to revise and resubmit our paper. We have gone through the manuscript thoroughly and corrected grammatical errors and further sought to clarify the text. In particular we have made changes to shorten and clarify the abstract text.

We have also added a Data Availability section at the end of the manuscript as requred. We have included comma delimited text with Supplemental Table 1 in the supplemental material. This may be rather cumbersome for readers. We have deposited the data in the Pangea respository, but unfortunately our doi did not yet get assigned, which is why I added the data to the supplemental file.

The figures should now all be at the appropriate resolution, and are included as a separate file.

Thank you so much for your patience in handling our paper, we know it was much improved by the review process.

Best regards,

Susan Trumbore (on behalf of co-authors)